# Assessing the performance of climate change simulation results from BESM-OA2.5 compared with a CMIP5 model ensemble

Vinicius Buscioli Capistrano[1,2], Paulo Nobre[1], Sandro F. Veiga[1], Renata Tedeschi[1], Josiane Silva[1], Marcus Bottino[1], Manoel Baptista da Silva Junior[1], Otacílio Leandro Menezes Neto[1], Silvio Nilo Figueroa[1], José Paulo Bonatti[1], Paulo Yoshio Kubota[1], Julio Pablo Reyes Fernandez[1], Emanuel Giarolla[1], Jessica Vial[3], and Carlos A. Nobre[4]

[1]Center for Weather Forecast and Climate Studies/National Institute for Space Research (CPTEC/INPE), Cachoeira Paulista – São Paulo, Brazil
[2]Amazonas State University (UEA), Manaus – Amazonas, Brazil.
[3]Laboratoire de Météorologie Dynamique/Centre National de la Recherche Scientifique (LMD/CNRS), Paris, France
[4]National Center for Monitoring and Early Warning of Natural Disasters (CEMADEN), São José dos Campos – São Paulo, Brazil

**Correspondence:** Vinicius Buscioli Capistrano, Amazonas State University, 1200 Darcy Vargas Ave., 69050-020, Manaus – Amazonas, Brazil (vcapistrano@uea.edu.br)

**Abstract.** The main features of climate change patterns, as simulated by the coupled ocean-atmosphere version 2.5 of the Brazilian Earth System Model (BESM), are compared with those of 25 other CMIP5 models, focusing on temperature, precipitation, atmospheric circulation and radiative feedbacks. The climate sensitivity to quadrupling the atmospheric $CO_2$ concentration was investigated via two methods: linear regression (Gregory et al., 2004) and radiative kernel (Soden and Held, 2006; Soden et al., 2008). Radiative kernels from both NCAR and GFDL were used to decompose the climate feedback responses of the CMIP5 models and BESM into different processes. By applying the linear regression method for equilibrium climate sensitivity (ECS) estimation, we obtained a BESM value close to the ensemble mean value. This study reveals that the BESM simulations yield zonally average feedbacks, as estimated from radiative kernels, that lie within the ensemble standard deviation. Exceptions were found in the high-latitudes of the Northern Hemisphere and over the ocean near Antarctica, where BESM showed values for lapse-rate, humidity feedbacks and albedo that were marginally outside of the standard deviation of the values from the CMIP5 multi-model ensemble. For those areas, BESM also featured a strong positive cloud feedback that appeared as an outlier compared with all analyzed models. However, BESM showed physically consistent changes in the temperature, precipitation and atmospheric circulation patterns relative to the CMIP5 ensemble mean.

# 1 Introduction

The effects of increased atmospheric $CO_2$ concentrations on the climate system have been studied over the last 120 years (Arrhenius, 1896; Callendar, 1938; Plass, 1956; Kaplan, 1960; Manabe and Wetherald, 1967, 1975; Manabe and Stouffer, 1980; IPCC, 2007, 2013; Pincus et al., 2016; Good et al., 2016, and many others). The human-induced increase in atmospheric greenhouse gas (GHG) concentrations, sometimes given as the $CO_2$-equivalent concentration, contributes to a radiation imbalance at the top-of-atmosphere (TOA) that causes less outgoing radiation to leave the Earth system. The trapping of infrared radiation results in a temperature rise at the lower levels of the troposphere, as well as an increase in ocean heat content. In addition, the increased GHG concentration can trigger climate feedback processes that either amplify or damp the initial radiative perturbation (Cubasch and Cess, 1990). Earth system models (ESMs) are the most advanced tools available for analyzing the coupled climate system (atmosphere, ocean, land, and ice) physical processes and their interactions, although even these models still exhibit important uncertainties in their projections of climate change (IPCC, 2013).

The equilibrium global-mean surface temperature change induced by doubling the $CO_2$ concentration in the atmosphere, referred to as the equilibrium climate sensitivity (ECS), remains a centrally important measure of a model's climate response to $CO_2$ forcing. In the fifth Intergovernmental Panel on Climate Change (IPCC) assessment report (AR5), climate model ECS estimates range from 2.0 K to 4.5 K. For more than 40 years, this inter-model spread has been considered one of the most critical uncertainties for the evaluation of future climate change (IPCC, 2013). This inter-model dispersion arises principally from differences in how the climate models simulate climate feedback processes. Among them, cloud feedback constitutes the largest source of variation in the climate sensitivity estimates (Cess et al., 1989, 1990; Dufresne and Bony, 2008; Vial et al., 2013; Caldwell et al., 2016).

Beyond the ECS, the response of the precipitation patterns to anthropogenic GHG emissions is a topic of great interest in climate science, given their potential effects on both societies and ecosystems. Changes in precipitation can generally be decomposed into two processes: a thermodynamic component due to increased moisture and no circulation change, and a dynamic component due to circulation change with no moisture change (Bony et al., 2006; Seager et al., 2010). The thermodynamic component gives rise to the well-known 'wet-gets-wetter' and 'dry-gets-drier' patterns of precipitation change first described by Held and Soden (2006), which are associated with the Clausius-Clapeyron relation (i.e., a temperature-dependent exponential increase in the saturation-specific humidity) (Marvel and Bonfils, 2013). The dynamic component, which is associated with circulation change, sometimes yields strong deviations from the thermodynamic pattern of precipitation, and this component is known to dominate the inter-model deviation in estimates of total precipitation due to uncertainties in the regional circulation change (Xie et al., 2015).

In this study, we assess the main features of climate change patterns as simulated by the Brazilian Earth System Model, ocean-atmosphere coupled version 2.5 (BESM-OA2.5), with a focus on temperature (climate sensitivity and feedbacks), precipitation and atmospheric circulation. The recent development of the BESM-OA2.5 has been a coordinated effort at the National Institute for Space Research (INPE) in Brazil intended to advance the understanding of the causes of global and regional climate changes and their effects on the socioeconomic sector. We evaluate how the BESM-simulated climate change

prediction compares with those from Coupled Model Intercomparison Project phase 5 (CMIP5) models, also discussing peculiarities in the BESM-OA2.5 climate response. This paper is structured as follows: section 2 presents the description of the new features of BESM-OA2.5, section 3 presents the methodology, section 4 presents the results, and section 5 presents the summary and conclusions.

## 2  Model Description

### 2.1  BESM-OA2.5

BESM-OA2.5 model is the result of coupling the Center for Weather Forecast and Climate Studies (CPTEC/INPE) Brazilian Atmospheric Model [BAM (Figueroa et al., 2016)] and the Geophysical Fluid Dynamics Laboratory (GFDL) Modular Ocean Model version 4p1 (Griffies et al., 2004) via the Flexible Modular System (FMS) (also from GFDL). The dynamical core and physical parameterisations of the atmospheric component of BESM-OA2.5 are the same as those discussed in Veiga et al. (2019). BAM is a hydrostatic model, with its dynamical core based on the spectral transform method, which employs global spherical harmonic basis functions. The Eulerian advection scheme option is used in this study, with a two-time-level semi-Lagrangian scheme for the transport of moisture and microphysics prognostic variables, which are carried out completely within the model grid space. Simplified fast physical parametrizations are used here to increase the computational efficiency of long integrations, thus resulting in a decreased computational demand compared with that required by the operational Numerical Weather Prediction (NWP) model. A summary of the main differences in the physical parameterizations between BAM (as used in this paper) and the BAM NWP operational model is provided in Table 1. The dynamical equations in BAM are discretized following a spectral transform with horizontal resolution truncated at triangular wavenumber 62 (an equivalent grid size of approximately 1.875°) and 28 layers unevenly spaced in the vertical sigma coordinate with the top level at approximately 2.73 hPa. The oceanic component uses a tripolar grid at a horizontal resolution of 1° in longitude, and in the latitudinal direction the grid spacing is 1/4° between 10°S-10°N, decreasing uniformly to 1° at 45° and to 2° at 90° in both hemispheres. The ocean grid has 50 vertical levels with a 10-m resolution in the upper 220 m, decreasing gradually to approximately 370 m at deeper levels.

Veiga et al. (2019) showed that BESM-OA2.5 can capture the general mean climate state; however, substantial biases appeared in the simulation associated with a double ITCZ over the Pacific and Atlantic Oceans and regional biases in the precipitation over the Amazon and Indian regions. BESM-OA2.5 can also reproduce the most important large-scale interannual and decadal climate variability patterns. The Atlantic Meridional Mode (AMM) (Nobre and Shukla, 1996) is well simulated by the model in terms of its spatial pattern and temporal variability, whereas this mode is poorly represented in most CMIP5 models (IPCC, 2013; Liu et al., 2013; Richter et al., 2014; Amaya et al., 2017). The maximum strength of the Atlantic Meridional Overturning Circulation (AMOC) represented by BESM-OA2.5 is 14 Sv, which is lower than the value determined within the RAPID project [17 Sv; McCarthy et al. (2015)] but in the range of the observed root-mean-square variability, and this value is comparable to the ensemble AMOC simulated by the CMIP5 models. Moreover, the spatial structures of both the North Atlantic Oscillation (NAO) and the Pacific Decadal Oscillation (PDO) variability are well captured (Veiga et al., 2019).

## 2.2 Comparison to a previous model version

The recently developed BESM-OA2.5 is an advancement of BESM-OA2.3, which was presented by Nobre et al. (2013). The main differences between BESM-OA2.5 and the previous version (BESM-OA2.3) lie in the atmospheric model and how some surface layer variables are estimated. The total energy balance at the TOA is better represented in BESM-OA2.5 than in BESM-OA2.3, with a reduced global mean bias of approximately -4 W m$^{-2}$, compared with -20 W m$^{-2}$ for the latter. It should be noted that BESM-OA2.5 has a new set of parameterizations, mainly related to an improved microphysical processes representation. For instance, the precipitation in the previous model was parameterized only in terms of the large-scale condensation. Moreover, BESM-OA2.5 underwent improvements in the representation of wind, humidity and temperature in the surface layer, with the use of the similarity functions method presented by Jiménez et al. (2012). Based on Monin-Obukhov theory, the wind ($u_{10m}$), humidity ($q_{2m}$) and temperature ($\theta_{2m}$) are estimated from the values of the first atmospheric model level and the surface, as described in Eq. (24), (25) and (26) of Jiménez et al. (2012). Furthermore, the similarity functions $\psi_m$ and $\psi_h$ depend on the stability regimes (Businger et al., 1971). For BESM-OA2.5, those regimes are associated with stable ($\zeta/L > 0$) and unstable ($\zeta/L \leq 0$) conditions (Arya, 1988). These diagnostic variables are important for BESM because they are used in the ocean-atmosphere coupling strategy.

Both versions reproduce the main climate variability, particularly over the Atlantic, as the AMOC and the AMM, but simulate a weak El Niño/Southern Oscillation (ENSO) interannual variability over the equatorial Pacific (Nobre et al., 2013; Veiga et al., 2019). Concerning the general mean present-day climate state, BESM-OA2.5 shows improvements in reproducing the Intertropical Convergence Zone (ITCZ), and it reduces both the global precipitation root-mean-square error (RMSE) and the SST RMSE compared with those modeled by BESM-OA2.3.

One-year-long global simulations and 6-hourly outputs were performed with BAM configured with surface layer schemes based on Arya (1988) and Jiménez et al. (2012), here called BAM-Arya (the original scheme) and BAM-Jimenez (the new scheme), respectively. The normalized RMSE was computed with respect to the reanalysis NCEP-DOE (National Centers for Environmental Prediction – Department of Energy) version 2 (Kanamitsu et al., 2002). The normalized RMSE of the wind at 10 m and the temperature and humidity at 2 m for the two surface layer schemes were investigated. Consistent improvements of BAM-Jimenez relative to BAM-Arya were noted in all three variables over the oceanic regions. The normalized RMSE analysis over the continents yielded less consistent results, with improved BAM-Jimenez representation of both winds and temperature, but with an inferior representation of the humidity field (figures not shown).

## 3 Methodology

### 3.1 Experimental design

For this study, climate simulations were performed using BESM-OA2.5 (hereinafter BESM) for the piControl (pre-industrial control scenario, run for 300 years with atmospheric $CO_2$ concentration invariant at 274 ppmv) and abrupt4xCO2 (run for 150 years after the abrupt quadrupling of atmospheric $CO_2$ at year 151 of the piControl simulation) scenarios; therefore, both

experiments were run in parallel for 150 years. These two scenarios are commonly employed in CMIP5 studies for climate sensitivity assessment (Taylor et al., 2012; Eyring et al., 2016). Climate change is evaluated as the difference between the abrupt4xCO2 and piControl experiments. In addition, BESM's results were compared with a selection of 25 CMIP5 models listed in Table 2. All models, including BESM, were interpolated at 2.5° x 2.5° longitude/latitude horizontal resolution. All CMIP5 model data are available from the Earth System Grid Federation (ESGF).

## 3.2 Climate change sensitivity estimates

Here we estimate the climate feedback using two different methods, using either a regression according to Gregory et al. (2004) or radiative kernels (Soden et al., 2004, 2008). The Gregory method is more straightforward computationally; however, it returns only a global-mean value. Moreover, the ECS can be estimated with this method. On the other hand, it is possible to obtain the seasonal feedback for every lat-lon point with the radiative kernel method; furthermore, the feedback can be decomposed into different processes.

### 3.2.1 Linear forcing-feedback regression analysis

The regression method for computing the thermal response to radiative forcing was applied for 26 CMIP5 models, including BESM. The method consists in the linear regression between the annual change (considering abrupt4xCO2 minus piControl) of the global-mean near-surface temperature ($\Delta T_{as}$) and the net radiation flux change ($\Delta R$) at the TOA.

If $G$ is the radiative forcing imposed on the climate system (here, associated with an abrupt increase in atmospheric $CO_2$ concentration) and $\Delta R$ is the resulting radiative imbalance in the global-mean net radiative budget at the TOA, then at any time, the response of the climate system to this radiative imbalance corresponds to the radiative forcing according to the following equation:

$$\Delta R = \lambda \Delta \overline{T}_{as} + G \tag{1}$$

where $\lambda$ ($< 0$) is the climate feedback parameter and $\Delta \overline{T}_{as}$ is the global-mean near-surface temperature change. In a sufficiently long simulation (coupled atmosphere-ocean models take millennia), the climate system reaches a new equilibrium when $\Delta R$ = 0. As $G$ can be approximated via backward regression towards $\Delta \overline{T}_{as}$=0, ECS can be estimated as ECS=$-G/\lambda$. As the ECS is the theoretical equilibrium temperature for doubling $CO_2$, in a quadrupling of $CO_2$ it is common to divide the result derived from 4xCO$_2$ simulations by 2 (Andrews et al., 2012).

By using this linear forcing-response framework, we can estimate climate sensitivity, radiative forcing, and feedback parameter following the method proposed by Gregory et al. (2004). The values of $\lambda$ (slope) and $G$ (y-intercept) are estimated via the ordinary least-square regression of the global-annual-mean of $\Delta R$ against $\Delta \overline{T}_{as}$ under all-sky conditions. Using the same linear technique, we decompose the feedback parameter into shortwave (SW) and longwave (LW) radiation components, and we extract the clear-sky radiative flux components from BESM and CMIP databases to estimate the cloud radiative forcing or cloud radiative effect ($\Delta$CRE) defined as the difference between the all-sky and clear-sky feedback parameters (Andrews et al., 2012). Estimates of $G$, $\lambda$, $\Delta$CRE, and ECS for all models are presented in the next section.

### 3.2.2 Separating individual climate feedbacks using radiative kernels

The radiative kernel technique [as in Soden and Held (2006), Soden et al. (2008), Vial et al. (2013)] is used to separate the feedback parameter $\lambda$ into contributions from the temperature response ($\lambda_T$), water vapor ($\lambda_{\ln q}$), surface albedo ($\lambda_a$), and cloud ($\lambda_c$) feedbacks plus a residual term (Re) (Vial et al., 2013) as expressed in Eq. (2).

$$\lambda = \lambda_T + \lambda_{\ln q} + \lambda_a + \lambda_c + \mathrm{Re} \tag{2}$$

We used both GFDL (Soden et al., 2008) and National Center for Atmospheric Research (NCAR) (Shell et al., 2008) radiative kernels to estimate climate feedbacks. Such radiative kernels represent the impact on the radiative balance at the TOA via arbitrary increases in the atmospheric temperature, albedo and specific humidity. For calculating the temperature kernel, an increment of 1-K is added for all model levels (including the surface). For the albedo kernel, the albedo value is increased by 0.01 (1%). Finally, for the water vapor kernel, the specific humidity is increased by a value equivalent to a 1-K atmosphere temperature increase, with the relative humidity remaining constant. Furthermore, we used the $\ln(q)$ instead of $q$, considering the quasi proportionality of the absorption of radiation by water vapor to $\ln(q)$.

Following Soden and Held (2006), Jonko et al. (2013) and Vial et al. (2013), we decompose the total feedback parameter ($\lambda$) into contributions from $\lambda_T$, $\lambda_{\ln q}$, $\lambda_a$, and $\lambda_c$ as:

$$\lambda = \sum_x \lambda_x + \mathrm{Re} = \sum_x \frac{\partial R}{\partial x}\frac{dx}{d\overline{T}_{as}} + \mathrm{Re} = \sum_x K_x \frac{dx}{d\overline{T}_{as}} + \mathrm{Re}$$
$$\lambda = \left( K_{T_s}\frac{dT_s}{d\overline{T}_{as}} + K_T\frac{dT}{d\overline{T}_{as}} \right) + \left( K_{\ln q}\frac{d\ln q}{d\overline{T}_{as}} \right)$$
$$+ \left( K_a\frac{da}{d\overline{T}_{as}} \right) + \lambda_c + \mathrm{Re} \tag{3}$$

The temperature feedback has been separated into the Planck feedback (the vertically uniform tropospheric warming equal to the surface warming) and the lapse rate feedback (the deviation from the tropospheric uniform warming):

$$\lambda_T = \lambda_p + \lambda_{lr} = \left( K_{T_s}\frac{dT_s}{d\overline{T}_{as}} + K_T\frac{dT_s}{d\overline{T}_{as}} \right)$$
$$+ \left( K_T\frac{dT}{d\overline{T}_{as}} - K_T\frac{dT_s}{d\overline{T}_{as}} \right) \tag{4}$$

In Eq. (3), $K_x$ (the radiative kernel for a variable $x$) and $x$ [temperature ($T_s$ and $T$, in K), the natural logarithm of humidity ($\ln q$, in kg/kg) and the albedo ($a$, dimensionless)] are functions of the longitude, latitude, and pressure vertical coordinates in the monthly climatology. To obtain tropospheric averages, the water vapor and temperature feedbacks are vertically integrated from the surface up to the tropopause, defined as 100 hPa at the Equator, and varying linearly to 300 hPa at the Poles. The stratospheric temperature and water changes is not accounted for in calculating the feedbacks, and they are shifted to the residuum.

It is worth noting that in the regression method, the radiative feedback is consistent with the actual radiative transfer scheme used in the climate model, while in the radiative kernel, the feedback is not necessarily consistent. In fact, the kernel is obtained

from another climate model that is not among the CMIP5 models analyzed. Model intercomparison is easily achieved via this method, as the same kernel can be applied to all models (Soden and Held, 2006; Soden et al., 2008). However, the resulting kernel derived feedbacks can only be assumed to reflect the actual feedback in the considered models under the premise of small differences between the radiative transfer codes.

Due to the non-linearities involving clouds and net radiation at the TOA (Soden et al., 2008), the cloud feedback is not calculated directly from these radiative kernels, which represents one of the key limitations of the kernel method. Instead, the cloud feedback is approximated using the cloud radiative forcing ($\Delta$CRE) corrected by removing the non-cloud feedbacks effect as in Soden et al. (2004, 2008). After the calculation of non-cloud feedbacks for both all-sky and clear-sky (superscript cs) conditions, we thus estimate the cloud feedback ($\lambda_c$) as:

$$\Delta\text{CRE} = \Delta R - \Delta R^{cs}$$

$$\Delta\text{CRE}_k = (G - G^{cs})_{CO_2} - \Delta\overline{T}_{as} \sum_x (\lambda_x - \lambda_x^{cs})$$

$$\Delta\text{CRE}_a = \Delta\text{CRE} - \Delta\text{CRE}_k$$

$$\lambda_c = \frac{\Delta\text{CRE}_a}{\overline{T}_{as}} \tag{5}$$

Where, $\Delta R^{cl}$ is the clear-sky net radiation flux at the TOA. Following Soden et al. (2008), $(G - G_{cl})_{CO_2}$ was defined as 2x0.69 W m$^{-2}$. The index "$k$" represents the change in the $\Delta$CRE due to the non-cloud feedbacks, while the index "$a$" means the adjusted $\Delta$CRE. Finally, a 30-year mean relative to the period from the 120th to 150th years of each scenario was used for all feedback estimations. It is worth mentioning that the term $\sum_x (\lambda_x - \lambda_x^{cs})$, introduced in Eq. 5, represents the cloud masking
on the non-cloud radiative feedbacks. It can be physically interpreted as the differences in the distribution of the temperature, vapor water and albedo between an all-sky and a clear-sky atmosphere (Soden et al., 2004).

### 3.3    Changes in the atmospheric circulation and precipitation

Monthly mean climatologies were computed for the last 30 years of the piControl and abrupt4xCO2 runs, and the projected climate response to $CO_2$ increase was evaluated from the difference between these abrupt4xCO2 and piControl monthly mean
climatologies. The statistical significance of this difference was calculated based on the Student t-test with a significance level of 90%. Furthermore, to evaluate how similar two spatial patterns are, we used the spatial inner product calculated as $\sum(A_i \cdot B_i)/(|A| \cdot |B|)$, where $A$ and $B$ are the 2-D variables and $i$ is the spatial index related to their lat-lon coordinates.

## 4    Results

### 4.1    $G$, $\lambda$, $\Delta$CRE and ECS estimated via the Gregory method

Figure 1 shows the linear regressions of $\Delta R$ , $\Delta$LW (clear-sky) and $\Delta$SW (clear-sky) against $\Delta\overline{T}_{as}$ from BESM. The linear regressions based on all-sky radiative flux are used to estimate ECS, $G$ and $\lambda$, while the regressions based on clear-sky data

are used to obtain $\Delta$CRE (as mentioned in the previous section). BESM features $G$ = 8.62 W m$^{-2}$, $\lambda$ = -1.45 W m$^{-2}$ K$^{-1}$, $\Delta$CRE = -0.13 W m$^{-2}$ K$^{-1}$, and ECS = 2.96 K.

The parameters $G$, $\lambda$, $\Delta$CRE and ECS as computed for all models are shown in Table 3. The climate sensitivities of the 26 CMIP5 coupled models (including BESM-OA2.5) were compiled in extension of the previous work by Andrews et al. (2012), who evaluated 15 CMIP5 coupled models. In the present work, we added the following models: ACCESS1-0, ACCESS1-3, bcc-csm1-1, BESM-OA2.5, BNU-ESM, CCSM4, FGOALS-g2, FGOALS-s2, GISS-E2-H, GISS-E2-R, and inmcm4. In Andrews et al. (2012) the ECS ranges from 2.07 to 4.74 K for the 15 models analyzed there, which is largely confirmed by our analysis. The small differences can possibly be attributed to the interpolation of the data. G and $\lambda$ vary from 5.01 to 8.95 W m$^{-2}$ and from -1.66 to -0.60 W m$^{-2}$ K$^{-1}$, respectively. The inter-model spread in G among the models is due to differences in the radiative codes used, as well as due to rapid adjustment processes in the troposphere and at the surface (Collins et al., 2006; Gregory and Webb, 2008; Andrews and Forster, 2008). The spread in the ECS is more robustly influenced by $\lambda$ than by $G$ (Figure 2), as was also suggested by Andrews et al. (2012). The correlation coefficient between the ECS and $\lambda$ is -0.82, which is significant at a 1% significance level (Figure 2b). On the other hand, the correlation between the ECS and $G$ is -0.01, which is not statistically significant (Figure 2a). Thus inter-model variation in the balance of feedbacks explains the dispersion in the ECS better than the initial radiative imbalance triggered by the $CO_2$ increase (related to $G$). Although BESM yielded one of the highest G values among all of the CMIP5 models, it showed a warming response to $CO_2$ doubling that is well within the range of 3.30$\pm$0.76 K as presented by the ensemble models.

The $\Delta$CRE from BESM is -0.13 W m$^{-2}$ K$^{-1}$, while the CMIP5 models yield $\Delta$CRE values ranging from -0.50 to 0.70 W m$^{-2}$ K$^{-1}$. Unlike the $\Delta$CRE$_a$, this term does not consider the masking effects of clouds as estimated via the radiative kernel method (Eq. 5). Therefore, the $\Delta$CRE cannot be interpreted to reflect a change in the cloud properties alone.

## 4.2 Climate feedbacks estimated via the radiative kernel method

Figure 3 shows the global-mean feedbacks for lapse-rate, water vapor, lapse-rate plus water-vapor, albedo, and cloud (SW, LW, and total) for each CMIP5 model. Both radiative kernels are used to test whether the results are sensitive to the choice of radiative kernel and whether the inter-model deviation is greater than the distribution of the radiatively active constituents (temperature, water vapor, albedo and cloud) of the base model. It is worth clarifying that positive/negative values of feedbacks contribute to the amplification/damping of global warming. The strongest positive feedback (Figure 3) is due to the water vapor (mean value: 1.39 W m$^{-2}$ K$^{-1}$), followed by clouds (mean value: 0.96 W m$^{-2}$ K$^{-1}$), and then surface albedo (mean value: 0.32 W m$^{-2}$ K$^{-1}$). The global mean Planck feedback is negative, with an average value of -3.60 W m$^{-2}$ K$^{-1}$ (not shown in Figure 3), followed by a lapse-rate feedback of -0.77 W m$^{-2}$ K$^{-1}$. BESM yields values near the ensemble mean for the albedo and cloud feedbacks, i.e., 0.27 W m$^{-2}$ K$^{-1}$ and 0.95 W m$^{-2}$ K$^{-1}$, respectively. For the lapse-rate feedback BESM yields a value of -0.71 W m$^{-2}$ K$^{-1}$, slightly underestimating the ensemble mean value in magnitude. In turn, BESM is among the models with the lowest global water vapor feedback average, with a value of around 1.24 W m$^{-2}$ K$^{-1}$.

Figure 4 shows the latitudinal profiles basic to the global mean feedback values of Figure 3, allowing to identify the regions that induce deviations of BESM results from the CMIP ensemble. In Figure 4a-b, there is a nearly constant Planck feedback

of approximately -3.4 W m$^{-2}$ K$^{-1}$ from 90°S to 60°N, with a notable increase in the ensemble standard deviation in the sub-Antarctic latitudes (around 60°S), which is in accordance with Rieger et al. (2017). The exception is in the Arctic region, where the mean value reaches -10 W m$^{-2}$ K$^{-1}$ with a similarly increased standard deviation value. In the sub-Antarctic and Arctic latitudes, BESM yields one of the most negative values for the Planck feedback, revealing that BESM has a stronger vertically

homogeneous warming (corresponding to large surface warming) among the CMIP5 models. Furthermore, for those same regions, BESM showed more positive lapse-rate feedback than the ensemble. Therefore, BESM yields a warming relatively larger at the surface and relatively weaker at the upper troposphere, resulting in a stronger vertical temperature gradient in comparison to the other models.

    In the Tropics, where there is an intense moist convection, atmospheric warming almost follows a moist adiabate (tem-

perature increase is larger in the upper troposphere compared to that at the surface), implying a negative lapse-rate feedback (Figure 4c-d) (Manabe and Stouffer, 1980). In accordance with this upper tropospheric warming in the Tropics, an increase in the specific humidity occurs (Manabe and Wetherald, 1975), which is causing a reinforcement of the greenhouse effect, reflected by a positive water vapor feedback as shown in Figure 4d-e. Because of this close link between the lapse-rate and water vapor feedbacks, it is common to consider their effects as the sum, as displayed in Figure 3. BESM shows a lapse-rate

feedback near the ensemble mean for the Tropics. The greatest BESM deviations are observed near the Antarctic and over the Arctic (Figure 4c-d), where this feedback became positive for all models. For the water vapor feedback, greater dispersion of the models was observed in the Tropics, with BESM systematically yielding values below the ensemble mean for the same latitude band (Figure 4e-f). These lower values extend throughout the Northern Hemisphere, consistent with the low global mean water vapor feedback value relative to the ensemble (shown in Figure 3).

The albedo feedback profiles from BESM and the CMIP5 models are compared in Figure 4g-h. Non-zero results mostly occur over the Polar regions, where there is a reduction in sea-ice and snow cover (Chung and Soden, 2015; Block et al., 2020). The positive albedo feedback signals yielded by all of the models in such regions imply that the reduction in the albedo corresponds to an increase in the radiation budget at the TOA, due to reduced upward shortwave radiation. As expected the Polar regions present a large dispersion among the models, which is related to how fast the sea-ice melts in the different climate

models. The regions over the Arctic and the ocean near the Antarctic show the largest surface warming, and this positive albedo, together with the positive lapse-rate feedbacks, are the main factor responsible for a phenomenon known as polar amplification (Pithan and Mauritsen, 2014; Block et al., 2020). BESM yielded an albedo feedback greater than the ensemble standard deviation over the Southern Ocean at around 60°S. This same latitude is where BESM shows negative Planck and positive lapse-rate feedbacks outside of the models limits, as previously discussed.

Finally, regarding cloud feedback, most of the inter-model spread arises from the SW component (Figures 3 and 4i-j). This dispersion is also reflected in the standard deviation and in the limit between the minimum and maximum of the zonally averaged cloud feedback shown in Figure 4i-j. The SW cloud feedback ranges from -0.28 to 1.40 W m$^{-2}$ K$^{-1}$, while the LW cloud effect ranges from 0.10 to 0.96 W m$^{-2}$ K$^{-1}$. The combined SW and LW cloud effects result in positive cloud feedback ranging from 0.35 to 1.69 W m$^{-2}$ K$^{-1}$. This result is similar to that found by Soden et al. (2008) for CMIP3 [IPCC AR4,

IPCC (2007)] models, who also reported a near-zero to positive cloud feedback. BESM presents positive values of around 0.5

W m$^{-2}$ K$^{-1}$ for both SW and LW cloud feedback, resulting in a total cloud feedback of about 1.0 W m$^{-2}$ K$^{-1}$ (as shown in Figure 3). The highest positive values are in regions with strong albedo feedback (Figure 4i-j).

Although BESM yielded global area-averaged feedbacks near the model ensemble mean values, differences are found mainly at high latitudes. In fact, for cloud feedback, BESM is an outlier due to a strong shortwave component response over both the Arctic and the Southern Ocean near the Antarctic. This effect is evident via the decomposition of the cloud feedback into the SW and LW components for the ensemble and BESM values, as shown in Figure 5. To assess the analytical causes of this strong BESM shortwave cloud feedback departure from the ensemble, we separately computed the contributors to the SW cloud feedback, i.e., the SW CRE [as described by Cess et al. (1989)] and the feedback cloud masks, as in Eq. (5). For the shortwave component, the feedback cloud masks are obtained for the albedo and SW water vapor feedbacks, performing the all-sky minus clear-sky radiation flux for each feedback. We find that the higher BESM cloud feedback values (Figure 6g-h) are mainly consequences of the sum of the SW CRE (Figure 6a-b) and the effect of cloud masking for the albedo feedback (Figure 6c-d) in the sub-Antarctic and Arctic regions. In turn, the cloud masking for the SW water vapor (Figure 6e-f) does not contribute to the higher positive BESM values. As shown in Figure 6, it is possible to attribute BESM's status as an outlier over the Arctic region to the SW CRE, while in the Southern Ocean (around 60°S), the major contribution comes from the albedo feedback cloud mask.

A deepened physical analysis to understand the BESM cloud feedback behavior in high latitudes is obtained by examining the zonal mean of the change in the cloud vertical profile for BESM, as shown in Figure 7. Over the Arctic and near the Antarctic, BESM showed an increase in the cloud fraction above 850 hPa and a decrease below that level, indicating an upward shift of low-level clouds (Figure 7a). However, the increase in cloud cover above 850 hP is stronger than the reduction below. Because of this increase in the total cloud fraction, a negative SW CRE (not adjusted) appears in those regions (Figure 6a-b), consistent with an increase in sun shading (Figure 7b). Moreover, the SW cooling is smaller than the heating provided by LW radiation due to the upward shift of the low-level clouds, as evident in the net effect (Figure 7d). Furthermore, the positive albedo and lapse-rate feedbacks (Figure 4c-f) are consistent with this vertical cloud shifting. In this manner, a loss of SW energy at the surface associated with an increase in the total cloud fraction explains the negative SW CRE, and the gain of LW energy is responsible for the sea-ice melting. Consequently, this gain of LW energy is indirectly linked to the albedo feedback cloud mask for BESM, since the mask $[\Delta a/\Delta \overline{T}_{as}(K_a - K_a^{cs})]$ is proportional to the albedo change ($\Delta a$). As discussed before, both the SW CRE and albedo feedback cloud mask contribute to the large positive cloud feedback over the Arctic and sub-Antarctic areas observed in BESM.

### 4.3 Changes in temperature, atmospheric circulation and precipitation

Figure 8 shows the annual mean surface temperature differences between the abrupt4xCO2 and piControl scenarios for the ensemble of 25 CMIP5 models and BESM. As clearly shown in Figure 8, despite the generalized increase in the surface temperature over most of the globe in both panels, BESM shows a generally lower temperature increase, principally over the continental areas. The CMIP5 ensemble yields a mean continental temperature increase of 6.78 K, while BESM yields a value of 5.57 K. Nevertheless, the spatial patterns of the temperature increases are similar, as measured by the spatial inner product

(as described in the previous section) between the upper two panels in Figure 8, which results in a value of 0.96 (values near 1 indicate that both variables have closely similar spatial pattern, whereas values near 0 mean that there is hardly any pattern correlation between variables). One point of interest within the scientific community is the relative low temperature increase over the subpolar North Atlantic, also referred as the warming hole (Drijfhout et al., 2012). In the CMIP5 ensemble mean, the North Atlantic does not show a temperature decrease, although it is the region with the smallest temperature increase globally, while BESM shows an area of temperature decrease in this region. Such a decrease is also present in 6 other analyzed models (CSIRO-Mk3-6-0, FGOALS-s2, GFDL-ESM2G, GFDL-ESM2M, GISS-E2-R, and inmcm4). These results are consistent with those of Drijfhout et al. (2012), who showed that both observations and CMIP5 models present maximum cooling in the center of the subpolar gyre. Those authors argue that there is evidence that both the subpolar gyre and the AMOC adjust in concert with different time lags.

The regions with the largest temperature increase in the abrupt4xCO2 scenario are the Polar regions, particularly over the Arctic. The equatorial Pacific shows a relative maximum in warming when the abrupt4xCO2 scenario is compared with the piControl, both in the CMIP5 ensemble and BESM. Such changes in the Pacific mean state are consistent with the IPCC-AR5, which reports that the Pacific Ocean becomes warmer near the equator compared to the subtropics in the CMIP5 projections (Liu et al., 2005; Gastineau and Soden, 2009; Cai et al., 2015). The scatter plot of the global average under the abrupt4xCO2 conditions versus the piControl conditions presented in Figure 8 provides additional information that helps to understand the dispersion around the mean value among the different models. Even though the outputs of most of the models lie in either quadrant 1 or 3 (top-right and bottom-left, respectively), it is not possible to claim any robust linear relationship. This result indicates that models with warmer/cooler mean climates in the piControl runs apparently do not show a corresponding warmer/cooler climate in the abrupt4xCO2 experiments. BESM yields a temperature near that of the ensemble in both the piControl and abrupt4xCO2 runs; consequently, it also showed a temperature increase near the ensemble mean, consistent with its Planck feedback (Figures 3 and 4a-b).

Figure 9 shows the precipitation changes between the abrupt4xCO2 and piControl scenarios for the multi-model ensemble and the BESM. The results are, in general, similar to those of Held and Soden (2006), with wet regions becoming wetter (near-equatorial and subpolar regions) and dry regions becoming drier (centered around 30° in both hemispheres). The precipitation pattern in the CMIP5 ensemble shows increased precipitation over the equatorial Pacific, which could be related to the equatorial Pacific warming pattern shown in the temperature change (Figure 8). Furthermore, the CMIP5 ensemble shows a decrease in precipitation in northern South America. The BESM precipitation pattern is similar to the spatial patterns in the CMIP5 ensemble, but with some notable exceptions. For example, the decreased precipitation over the South Pacific shown in the CMIP5 ensemble plot is extended into the Indonesian region in BESM. It is also worth noting that in the BESM simulation, the South Pacific convergence zone (SPCZ) shifts southward in the abrupt4xCO2 scenario, compared with its position in the piControl scenario. In both the multi-model ensemble and BESM, the precipitation change pattern over South America is similar to that which occurs during El Niño years (Kayano et al., 1988; Marengo and Hastenrath, 1993; Grimm and Tedeschi, 2009), with increased precipitation over southeastern South America and decreased precipitation over northern/northeastern South America. The scatter plot in Figure 9 emphasizes a linear relationship between the experiments, indicating that models with higher

(lower) global average precipitation in the piControl scenario show higher (lower) precipitation in the abrupt4xCO2 scenario. As obvious from Figure 9, the BESM performance perfectly matches the ensemble mean behavior in global mean.

Figure 10 shows a scatter plot of the ECS versus the precipitation difference between the Abrupt4xCO2 and piControl scenarios ($\Delta Pr$) for all of the considered models. It is worth noting that all the models show increased global-mean precipitation upon quadrupling of atmospheric $CO_2$ with piControl pre-industrial $CO_2$ concentrations (positive values in y-axis in Figure 10). An apparent linear relationship between these differences (abrupt4xCO2 minus piControl) in the global-mean precipitation and ECS is also evident in Figure 10, in which the warmest models tend to have the largest precipitation changes. The slope of the linear regression reflects a 2.5% precipitation change per K, which is close to that found by Held and Soden (2006). This slope is much lower than that predicted by the Clausius-Clapeyron relation, i.e., an approximately 6.5% change in precipitation per K. In fact, such precipitation increases are not governed by moisture availability but rather by the surface and tropospheric energy balance, which incorporates the surface radiative heating, surface latent heat flux and radiative cooling of the troposphere (Allen and Ingram, 2002).

MRI-CGCM3, ACCESS1-0, and HadGEM2-ES show greater deviations from the linear fit shown in Figure 10. Furthermore, BESM is marginally out of the residual standard error interval, with a 9.5% increase in precipitation (the error limit is 9.2%). ACCESS1-0 and HadGEM2-ES use the same atmospheric model (Bi et al., 2013; Dix et al., 2013), which could explain the lower increase in precipitation in both coupled models.

In addition to temperature and precipitation changes, we are also interested in understanding the changes in the BESM atmospheric circulation (compared to other models) following a quadrupling of the $CO_2$ concentration. The sea level pressure (SLP) response patterns shown in Figure 11 depict a poleward shift in the subtropical high-pressure cells in both the CMIP5 ensemble and BESM. Furthermore, when the models are subjected to the increased atmospheric $CO_2$ concentration, a decrease in the SLP over the Polar regions is evident. This, connected with the increase in the mid-latitudes, indicates a positive trend in Arctic oscillation (AO) and Antarctic oscillation (AAO) episodes, which have already been reported by Fyfe et al. (1999), Cai et al. (2003), and Miller et al. (2006). It is also interesting to note the statistically significant SLP decrease (increase) over the eastern (western) Pacific, a pattern that might indicate an ENSO-like pattern in scenarios with an increased $CO_2$ concentration. This pattern is consistent with those depicted in Figure 8 for the SST changes in a 4xCO2 scenario.

The results from the piControl scenario (the contours in Figure 12) show that the Southern Hemisphere subtropical jet, reflected by the core of the maximum eastward zonal wind, is localized around $35°S$, 200-150 hPa in both the CMIP5 ensemble and BESM. In both panels of Figure 12 (BESM and the CMIP5 ensemble), we note that the regions with the strongest increases in westerly winds at all levels show a southward jet displacement. This observation is consistent with the poleward displacement of the high SLP center shown in Figure 11. Furthermore, as the high-pressure centers experienced a poleward shift, the pressure gradients intensified in the subpolar areas, consequently, the near-surface wind velocity increased, following the geostrophic approximation [$u \approx -(1/f\rho)(\partial p/\partial y)$], where $f$ is the Coriolis parameter and $\rho$ is the air density.

Figure 13 shows the average differences from $5°N - 5°S$ (Walker circulation) between the abrupt4xCO2 and piControl scenarios for vertical motion (omega, in shades) and zonal wind and vertical motion (vectors). According to the vertical motion pattern in the piControl (contours), the multi-model ensemble and BESM show subsidence over an extensive area in the Pacific

(150°E – 90°W) whose intensity is lower in the abrupt4xCO2 simulation, as shown in Figure 13 (blue). This finding is coherent with near-surface temperature patterns (Figure 8), which show an equatorial warming pattern in the mean state (e.g., during El Niño years, a weakening of the Walker circulation occurs). Furthermore, there are enhanced subsidence for the difference between the two scenarios over South America (around 75°W), consistent with the decrease in precipitation in tropical South America, in both BESM and the CMIP5 ensemble (Figure 9).

## 5 Conclusions

The piControl and abrupt4xCO2 scenarios for 25 CMIP5 models have been compared with those generated by the BESM model, based on their key sensitivity parameters, such as the equilibrium climate sensitivity (ECS) and climate feedbacks. Furthermore, changes in the temperature, atmospheric circulation and precipitation patterns were investigated.

Applying the linear regression method (Gregory et al., 2004), we obtained the ECS values for the 25 CMIP5 models analyzed, which ranged from 2.07 to 4.74 K, with BESM showing 2.96 K, close to the ensemble mean value ($3.30 \pm 0.76$). BESM has one of the biggest radiative forcing (G) values, i.e., $8.62$ W m$^{-2}$ K$^{-1}$, which is related to the radiative transfer model and the rapid adjustment process (Collins et al., 2006; Gregory and Webb, 2008; Andrews and Forster, 2008). Both G and the climate sensitivity ($\lambda$) define the ECS values calculated with this method; however, only $\lambda$ shows a significant correlation with the ECS, corroborating the results of Andrews et al. (2012).

To go further in the analysis, the radiative kernel method was used to separate the climate feedback into Planck, lapse-rate, water vapor, albedo and cloud feedbacks. Two regions presented considerable inter-model variability for the Planck, lapse-rate and albedo values, i.e., the Arctic region and over the ocean near the Antarctic. Over these regions, the BESM zonal mean cloud feedback ranges outside the standard deviation for the analyzed models, reaching approximately 3 W m$^{-2}$ K$^{-1}$, while the zonal mean was close to zero. BESM showed an upward shift of the low-cloud cover and an increase in cloud cover between 850 and 700 hPa, and these features were responsible for sun shading at the surface, which increased the reflected solar radiation at the TOA. Moreover, BESM presented a greater albedo change compared with those of the other models, especially in the sub-Antarctic area. Despite of the loss of SW energy at surface, which results in a negative SW cloud radiative effect, this effect was overcome by the albedo feedback cloud mask, which contributes to positive cloud feedback over those regions.

The atmospheric circulation patterns in BESM were similar to the patterns of the multi-model ensemble and those of other studies regarding the near-surface temperature (IPCC, 2007, 2013). For precipitation, the thermodynamic component reflects the well-known 'wet-gets-wetter' and 'dry-gets-drier' patterns of precipitation change (Held and Soden, 2006). BESM as well as the CMIP5 ensemble show consistent weakening of the Walker circulation, principally in the Pacific and over northern South America, which has been reported in previous studies (Collins et al., 2010; DiNezio et al., 2012; Huang and Xie, 2015; Cai et al., 2015). Regarding SLP, both BESM and the CMIP5 ensemble indicate a poleward displacement of the subtropical high-pressure systems, as shown in other studies (Fyfe et al., 1999; Cai et al., 2003; Miller et al., 2006). In line with such displacement, the subtropical jet also shifted polewards, and this effect was more distinct in the Southern Hemisphere.

Summarizing, we conclude that BESM-OA2.5 is a climate model that can reproduce approved physical processes that determine and modify changes of the global climate system. In this sense, the analysis methods used here have the potential to explain remaining process uncertainties causing inter-model spread in the cloud feedback in future work. Notwithstanding, the BESM team continues is effort to improve the cloud parameterization of the model as well as its land surface model in

5    subsequent versions. Furthermore, it is important to mention that the radiative energy imbalance of -4 W m$^{-2}$ at the TOA, arising from our ocean-atmosphere coupling, is seen as an issue that is to be tackled in ongoing model development work. We hope that the next version will include improved energy flows diagnostics and that it will include physical parameterizations of atmosphere/ocean interaction that lead to better agreement with other models and with observations.

**Code and data availability**

10   The BESM-OA2.5 source code is freely available subject to a license agreement. Please contact Paulo Nobre to obtain the BESM-OA2.5 source code and data.

*Competing interests.* The authors declare that they have no conflict of interest.

*Acknowledgements.* This work was supported by the Sao Paulo Research Foundation (FAPESP, 2009/50528-6, 2014/50848-9 and 2018/06204-0), the National Coordination for High Level Education and Training (CAPES, Grant 16/2014), the CAPES/National Water Agency (CAPES/ANA,

15   88887.115872/2015-01), the Brazilian National Council for Scientific and Technological Development (CNPq, 490237/2011-8 and 302218/2016-5), the Brazilian Research Network on Global Climate Change (FINEP/Rede Clima, Grant 01.13.0353-00), the National Institute for Science and Technology on Climate Change (INCT-MC 573797/2008-0), and INCT-MC Phase 2 funded by CNPq (Grant 465501/2014-1).

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

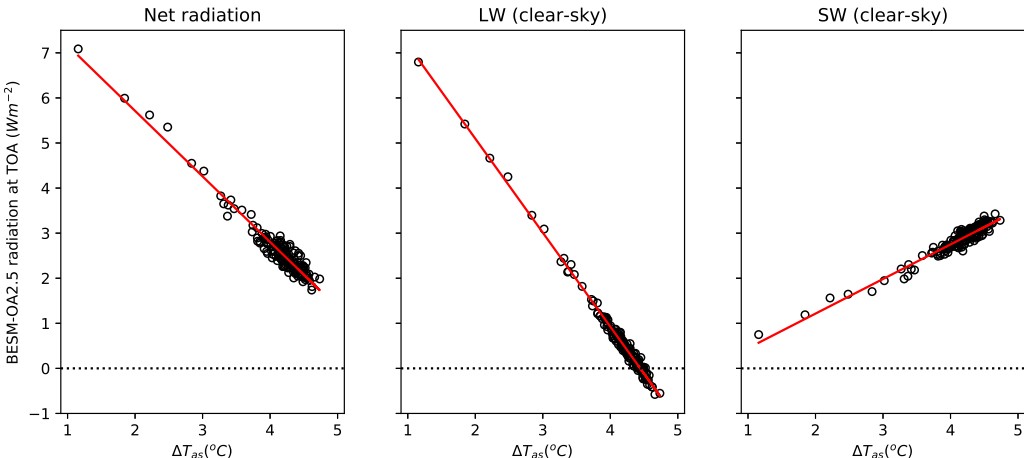

**Figure 1.** Annual global-mean linear regression between $\Delta \overline{T}_{as}$ and: (a) Net radiation, (b) $\Delta$LW (clear-sky), and (c) $\Delta$SW (clear-sky) for BESM-OA2.5

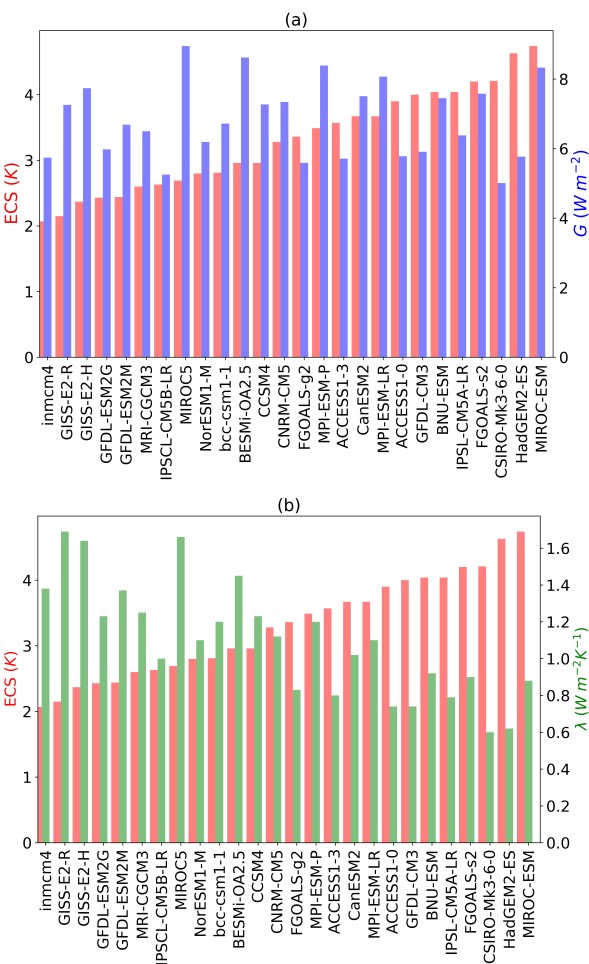

**Figure 2.** (a) Equilibrium climate sensitivity (ECS, in red) and radiative forcing ($G$, in blue) values with ECS values increasing from left to right; (b) ECS (red) and climate sensitivity ($\lambda$, in green) with ECS values increasing from left to right.

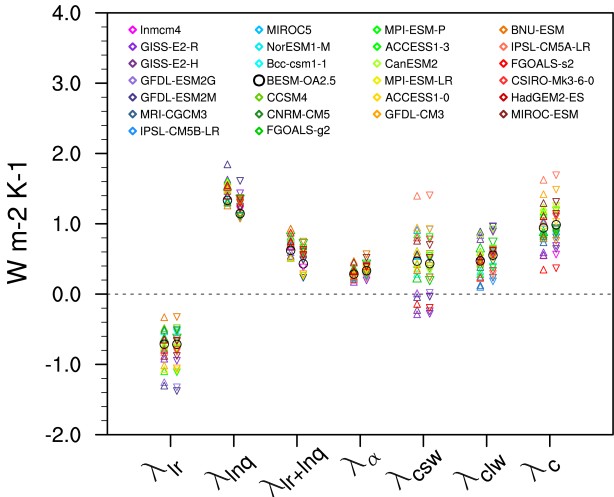

**Figure 3.** Global-mean feedbacks for 25 CMIP5 models and BESM-OA2.5 (circle). Changes in abrupt4xCO2 relative to the piControl were averaged over years 120-150. The triangles represent the mean estimated feedback values calculated using the NCAR radiative kernel whereas the upside-down triangles represent the estimated feedback values calculated using the GFDL radiative kernel.

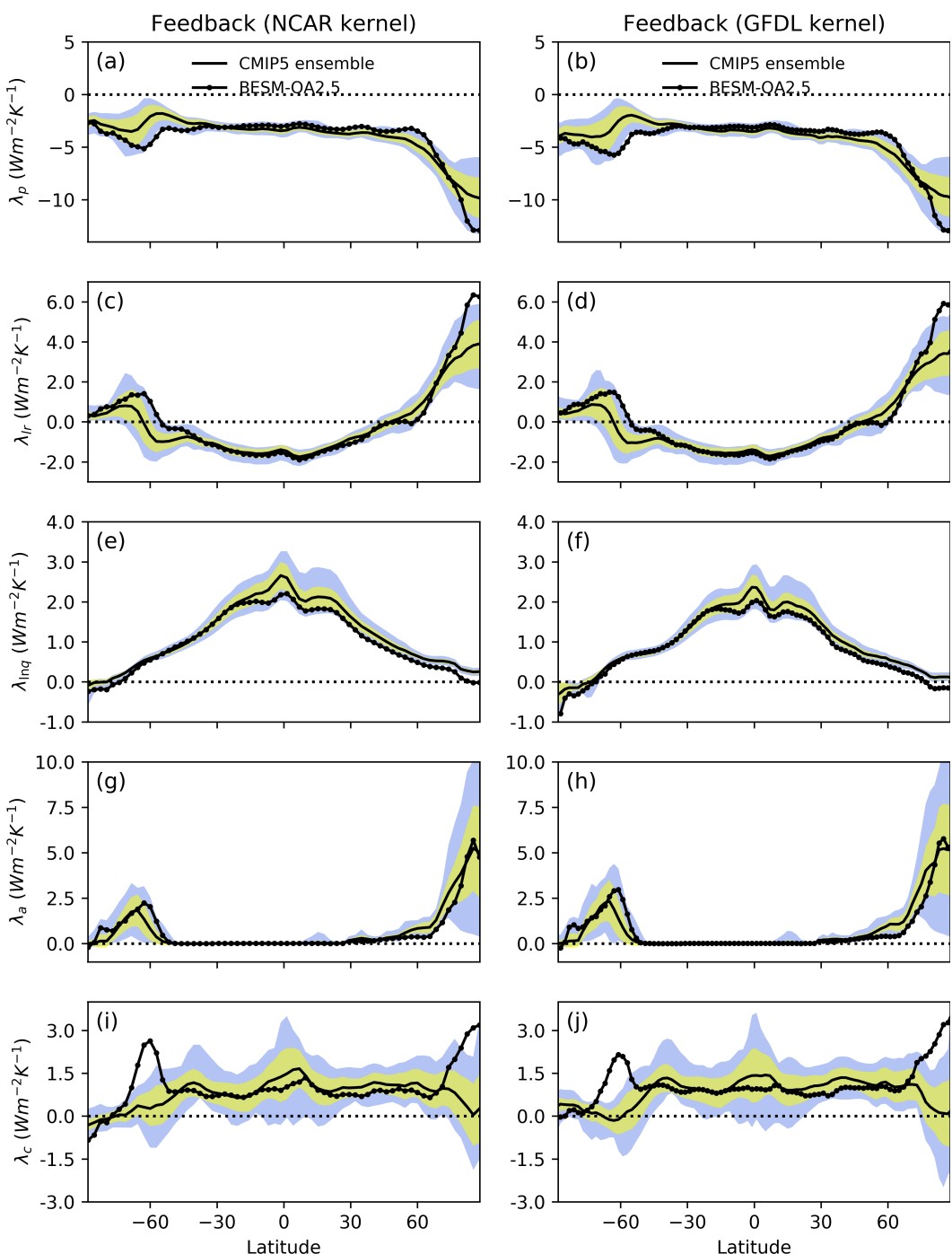

**Figure 4.** Feedbacks for the CMIP5 multi-model ensemble-mean (solid line) and BESM-OA2.5 (solid line with dots) for the Planck feedback (a and b), lapse-rate feedback (c and d), water vapor feedback (e and f), albedo feedback (g and h), and cloud feedback (i and j). Inter-model standard deviations for each latitude are in yellow. In blue are the feedback limits based on the maximum and minimum values for each latitude among the models, excluding BESM-OA2.5. All feedbacks are based on the averaged over years 120-150.

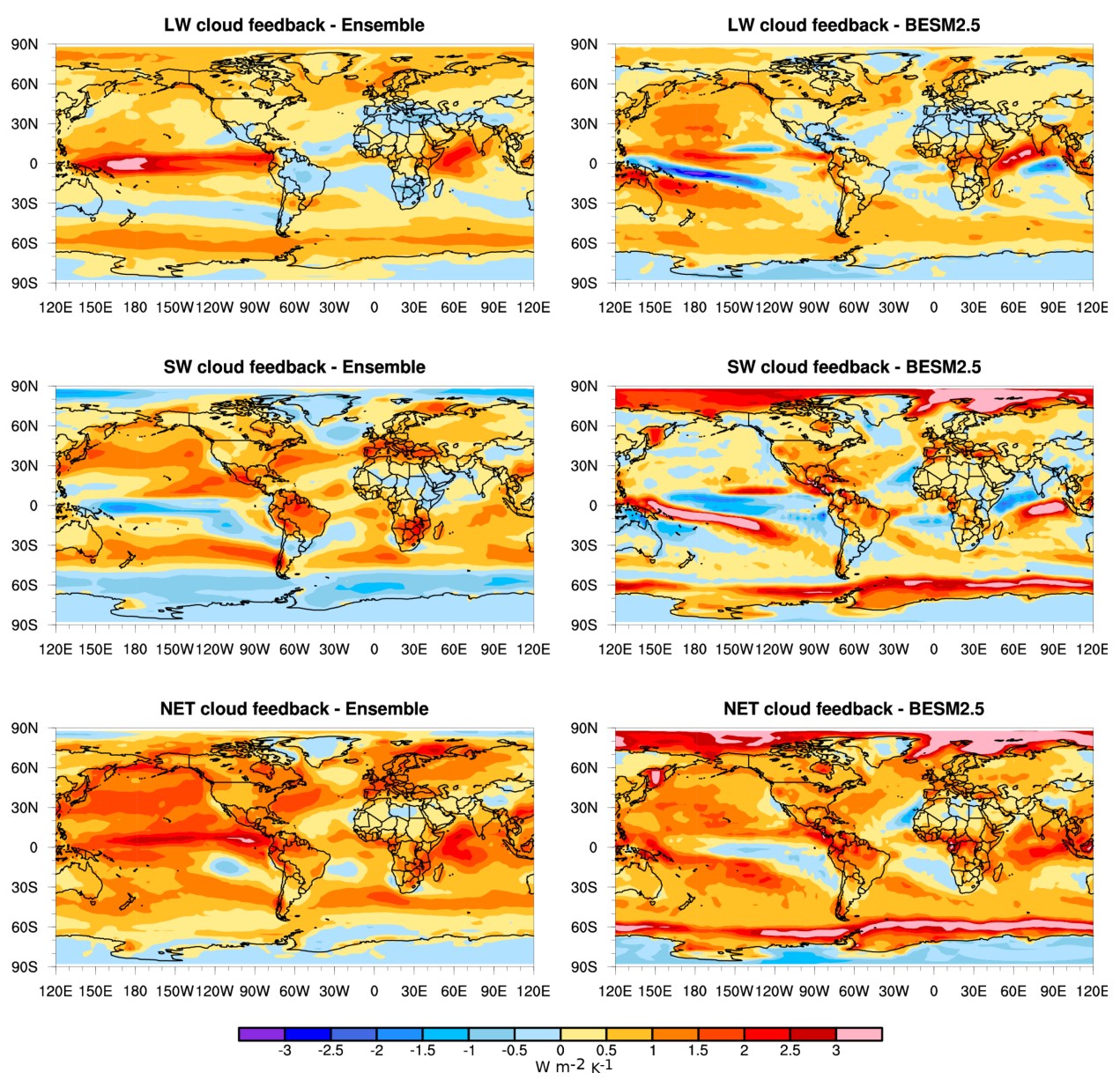

**Figure 5.** Cloud feedbacks calculated using the NCAR radiative kernel for the CMIP5 ensemble (left column) and BESM-OA2.5 (right column). These results are based on the averaged from years 120-150.

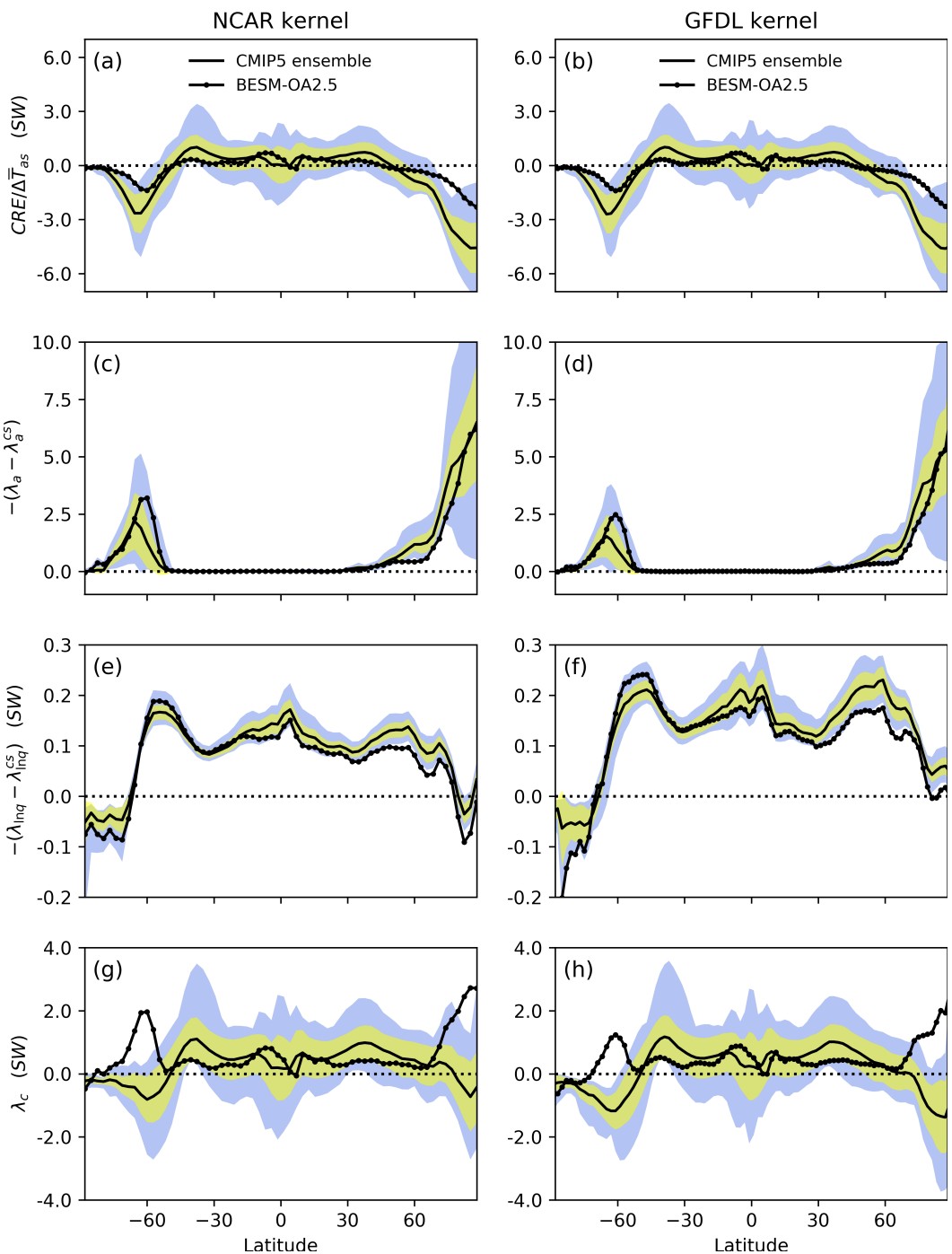

**Figure 6.** Shortwave cloud radiative effect (a and b), the albedo (c and d) and shortwave water vapor (e and f) feedbacks cloud masking, and shortwave cloud feedback (g and h) for the CMIP5 multi-model ensemble-mean (solid line) and BESM-OA2.5 (solid line with dots). Inter-model standard deviations for each latitude are in yellow. In blue are the feedback limits based on the maximum and minimum values for each latitude among the models, excluding BESM-OA2.5. The physical units of these feedbacks are W m$^{-2}$ K$^{-1}$.

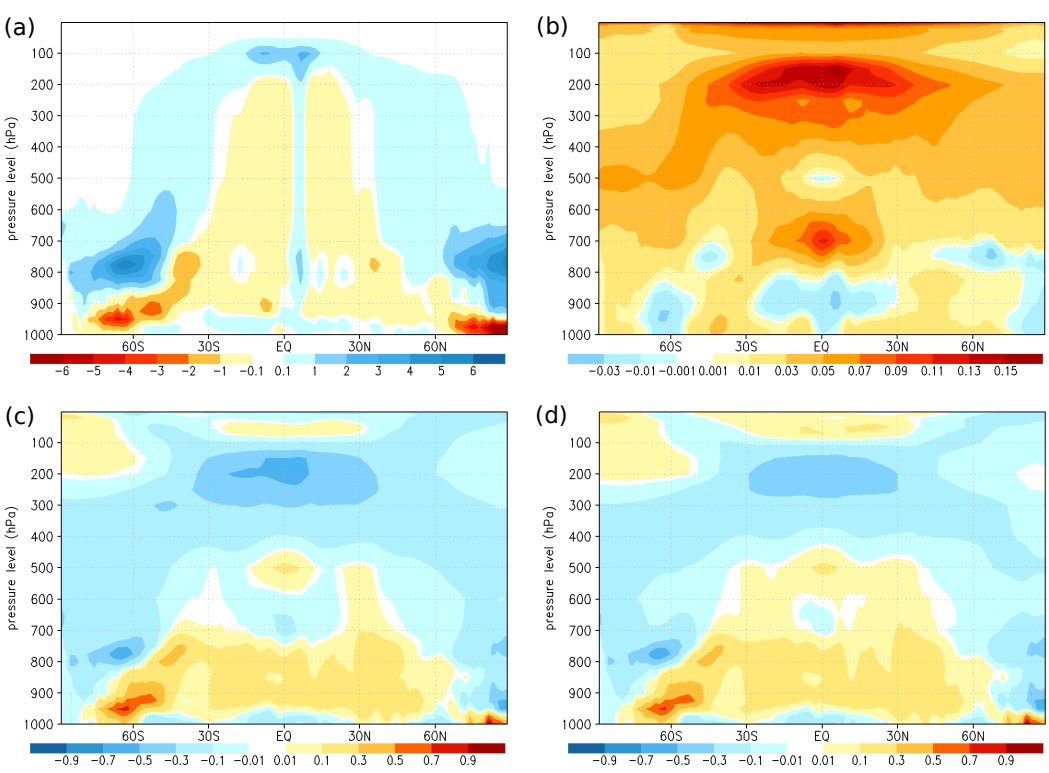

**Figure 7.** Vertical profiles of the zonal mean of the 4xC$_2$ - piControl mean difference for the following variables: (a) cloud fraction, radiative heating/cooling rate (dT/dt, in K day$^{-1}$) of (b) shortwave, (c) long wave and (d) sum of the shortwave and longwave for BESM-OA2.5.

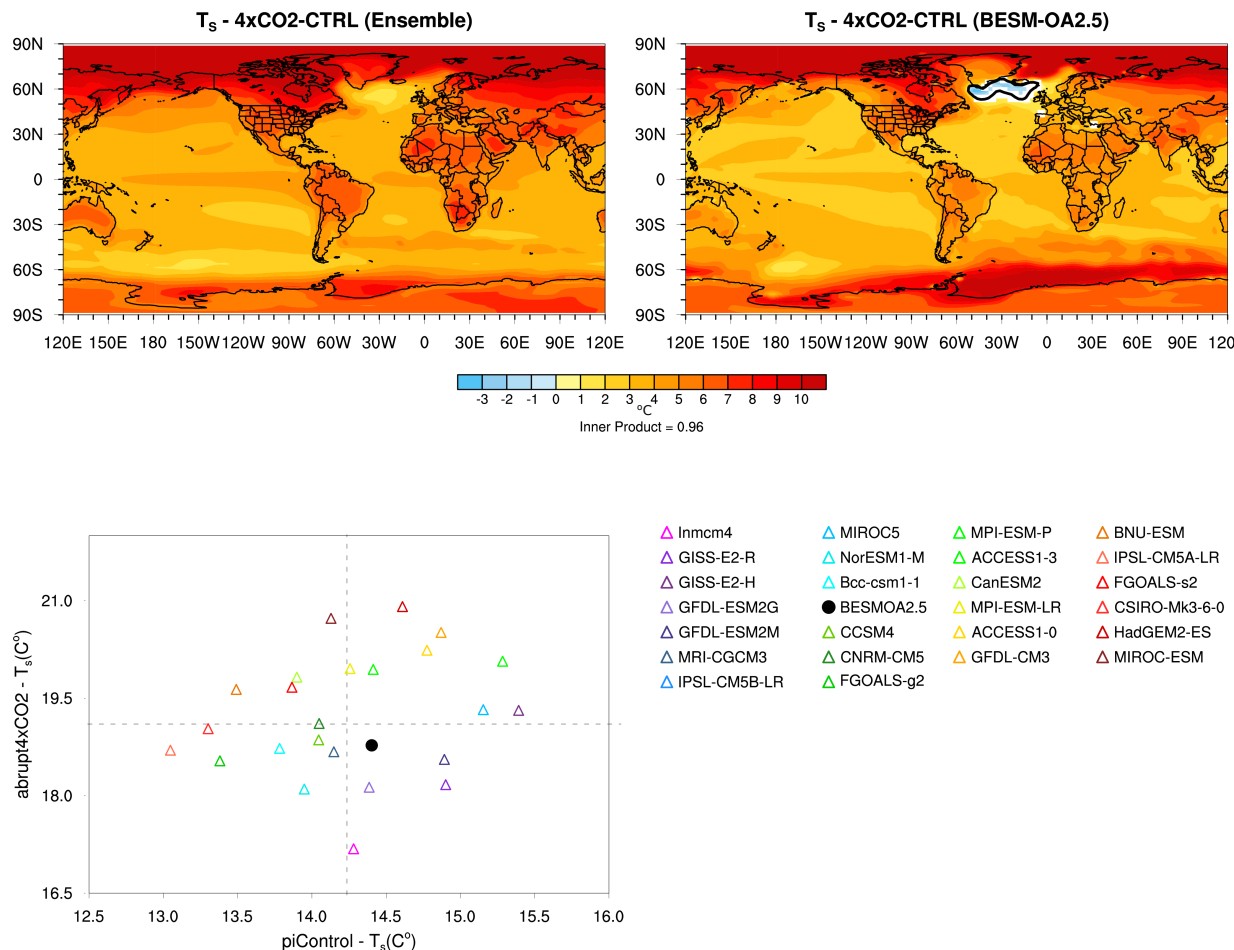

**Figure 8.** Differences (averaged over years 120-150) in surface temperature between the abrupt4xCO2 and piControl simulations in (a) the CMIP5 ensemble and (b) BESM-OA2.5. (c) A scatter plot of the global average surface temperature for the CMIP5 models used in the ensemble and BESM-OA2.5 (black dot). The shaded areas in (a) and (b) have confidence level greater than 90%; the black line represents the contour line of zero temperature difference.

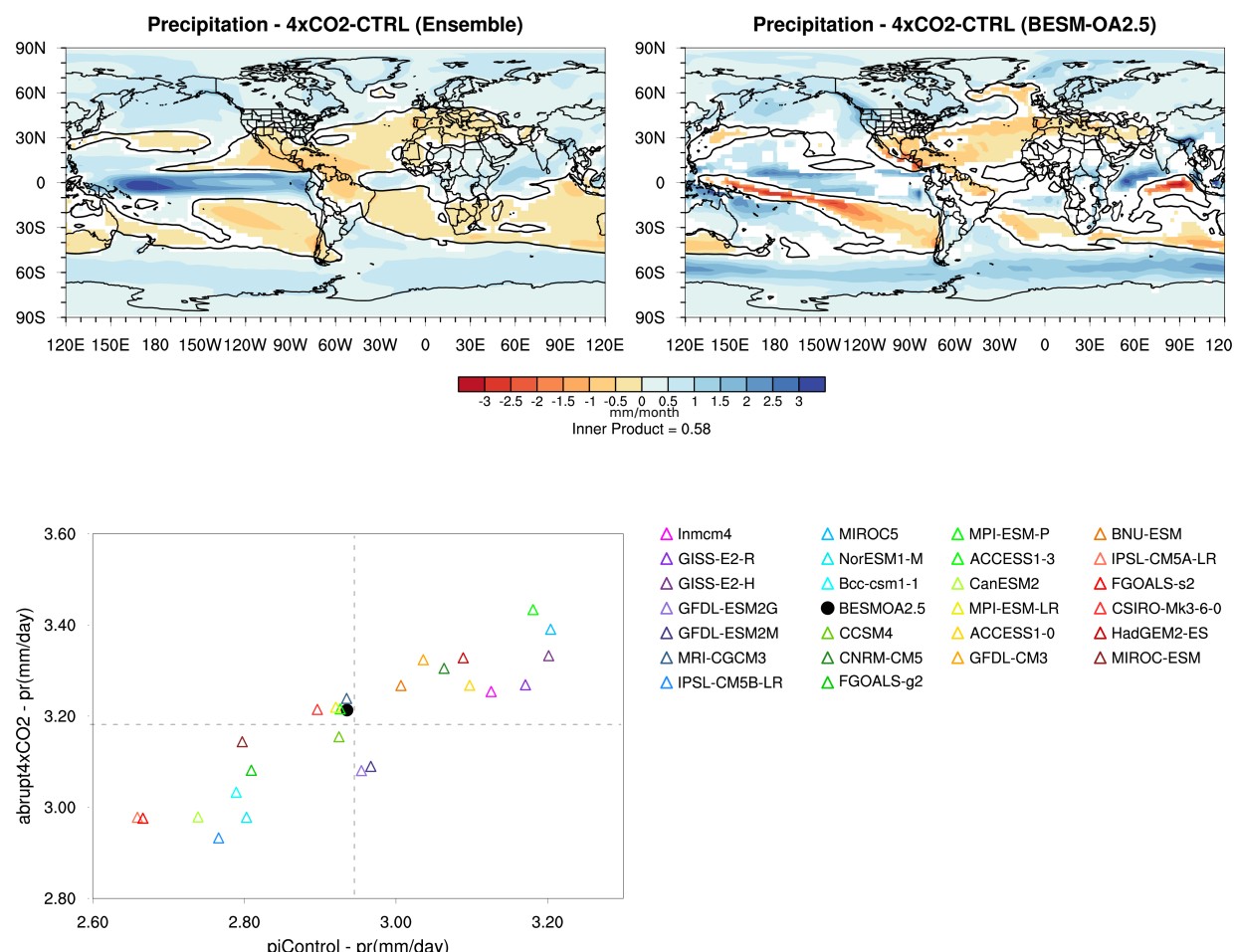

**Figure 9.** Differences (averaged over years 120-150) in precipitation (in mm/month) between the abrupt4xCO2 and the piControl simulations in (a) CMIP5 ensemble and (b) BESM-OA2.5 (c) A scatter plot of the precipitation global averages for the CMIP5 models used in the ensemble and BESM-OA2.5 (black dot). The shaded areas in (a) and (b) have confidence levels greater than 90%; the black line represents the contour line of zero precipitation difference.

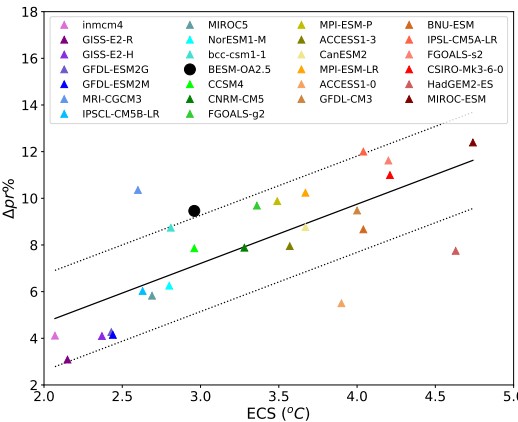

**Figure 10.** Scatter plot of the ECS and $\Delta Pr(\%)$ values for all of the ensemble models. The solid black line shows the linear fit between the ECS and the perceptual precipitation change. As in Figure 2, the models are sorted according their ECS value. The dash lines represent the error limits considering the residual standard error.

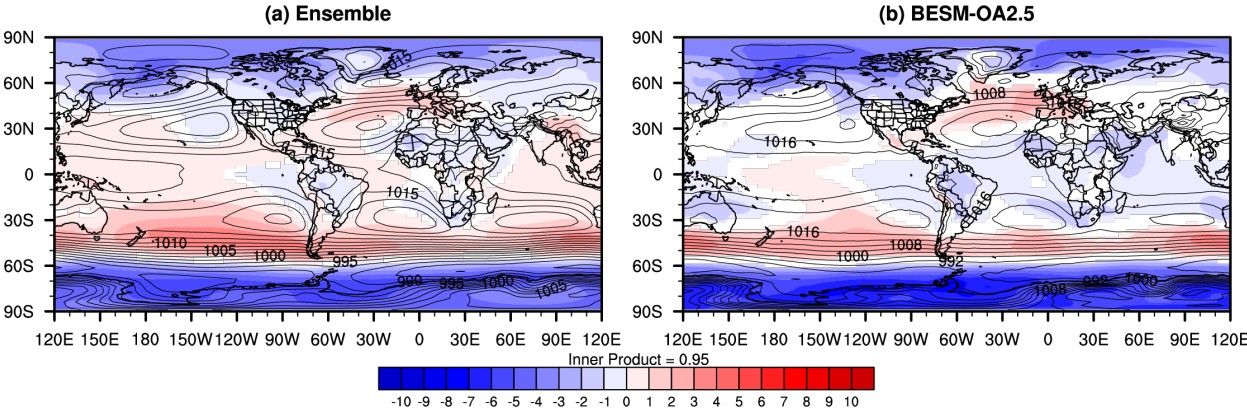

**Figure 11.** Difference (averaged over years 120-150) in sea level pressure (SLP) in hPa between two scenarios (abrupt4xCO2 minus piControl, shaded), and SLP under piControl conditions (contours) in CMIP5 models ensemble (first column) and BESM-OA2.5 (second column). The white areas have confidence levels less than 90%.

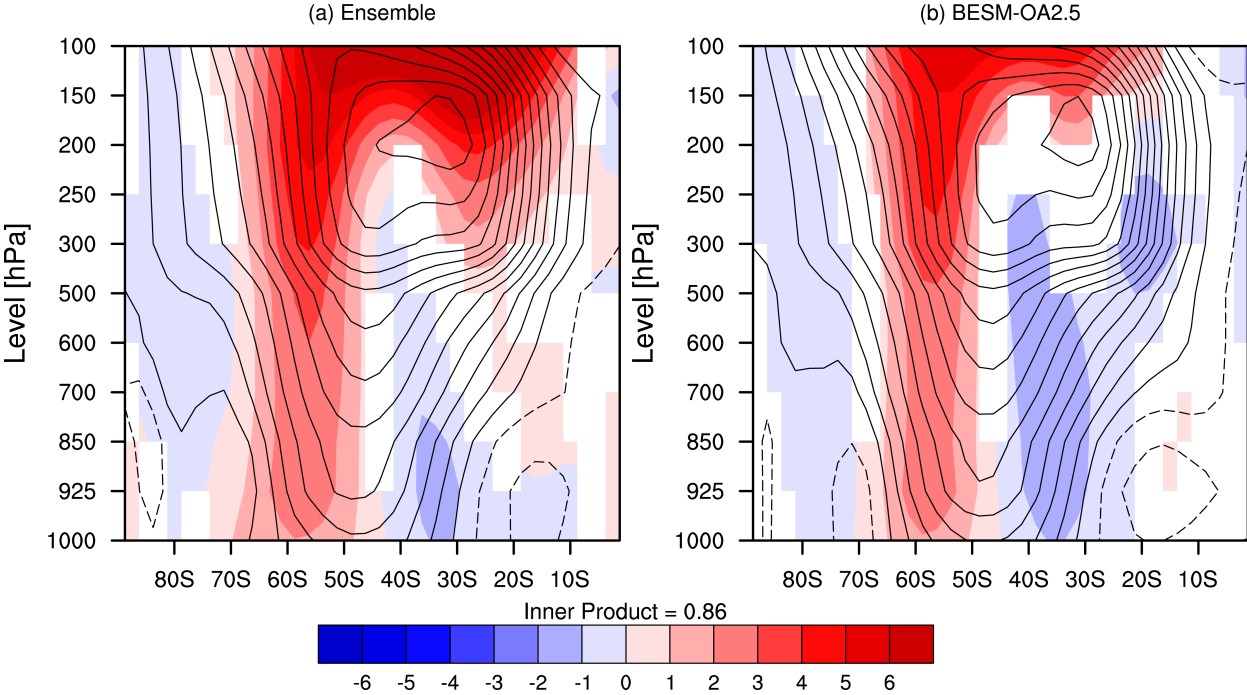

**Figure 12.** Vertical profile of the difference (averaged over years 120-150) in the zonal wind (in m/s) between two scenarios (abrupt4xCO2 minus piControl, shaded), and the piControl conditions (contours) for (a) the ensemble of CMIP5 models and (b) BESM-OA2.5. The white areas have confidence levels less than 90%.

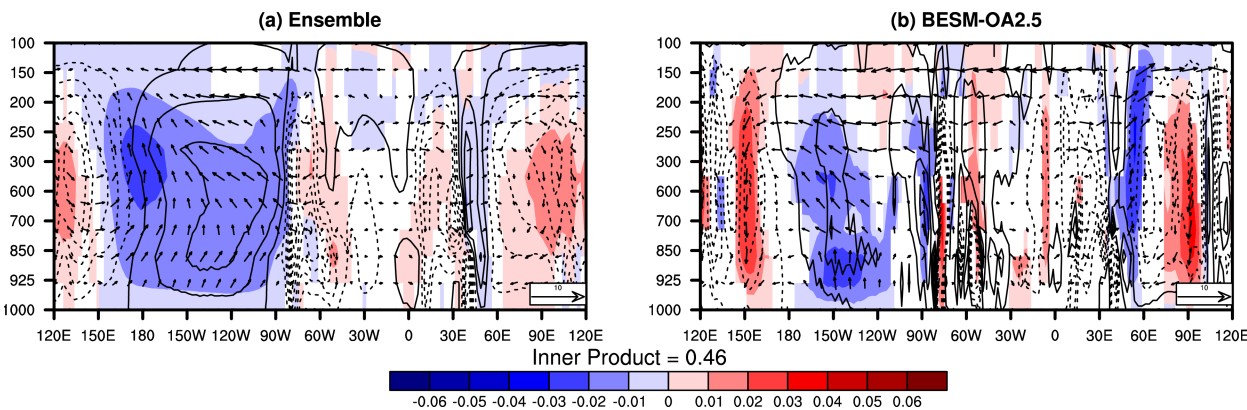

**Figure 13.** Difference (averaged over years 120-150) between the abrupt4xCO2 and piControl conditions for generalized vertical motion [omega (Pa/s), in shades] and the mean zonal and vertical motion (vectors), averaged between 5°S and 5°N, for (a) the CMIP5 ensemble and (b) BESM-OA2.5. The contours represents the averaged piControl vertical motion (omega) in the same region. The white regions have confidence levels less than 90%.

**Table 1.** Atmospheric physical parameterizations used in BAM (Figueroa et al., 2016) BESM-OA2.5.

| Physical Parameterization | BAM | BESM-OA2.5 |
|---|---|---|
| Shortwave radiation | RRTMG (Iacono et al., 2008) | Clirad (Tarasova and Fomin, 2007) |
| Longwave radiation | RRTMG (Iacono et al., 2008) | Harshvardhan et al. (1987) |
| Cloud microphysics | Morrison (Morrison et al., 2005) | Ferrier et al. (2002) |
| Land surface model | Ibis [Foley et al. (1996) modified by Kubota (2012)] | SSib (Xue et al., 1991) |
| Planetary Boundary Layer | Modified Mellor and Yamada (1982) scheme | Holtslag and Boville (1993) scheme |
| Shallow Convection | UW shallow convection (Park and Bretherton, 2009) | Tiedtke (1984) |
| Deep Convection | Modified Grell and Dévényi (2002) ensemble scheme | Modified Grell and Dévényi (2002) ensemble scheme |
| Gravity wave | Webster et al. (2003) scheme with low-level blocking | Alpert et al. (1988) |
| Total Cloud cover fraction | Based on Probability Density Function (PDF) | Slingo (1987) |

**Table 2.** Models belonging to the CMIP5 ensemble used in this study.

| Number | Model | Institution, country |
|---|---|---|
| 1 | ACCESS1-0 | CSIRO-BOM, Australia |
| 2 | ACCESS1-3 | |
| 3 | bcc-csm1-1 | BCC, China |
| 4 | BNU-ESM | BNU, China |
| 5 | CanESM2 | CCCma, Canada |
| 6 | CCSM4 | NCAR, USA |
| 7 | CNRM-CM5 | CNRM-CERFACS, France |
| 8 | CSIRO-Mk3-6-0 | CSIRO-QCCCE, Australia |
| 9 | FGOALS-g2 | LASG-CESS, China |
| 10 | FGOALS-s2 | LASG-IAP, China |
| 11 | GFDL-CM3 | |
| 12 | GFDL-ESM2G | NOAA-GFDL, USA |
| 13 | GFDL-ESM2M | |
| 14 | GISS-E2-H | NASA-GISS, USA |
| 15 | GISS-E2-R | |
| 16 | HadGEM2-ES | MOHC, England |
| 17 | inmcm4 | INM, Russia |
| 18 | IPSL-CM5A-LR | IPSL, France |
| 19 | IPSL-CM5B-LR | |
| 20 | MIROC-ESM | MIROC, Japan |
| 21 | MIROC5 | |
| 22 | MPI-ESM-LR | MPI-M, Germany |
| 23 | MPI-ESM-P | |
| 24 | MRI-CGCM3 | MRI, Japan |
| 25 | NorESM1-M | NCC, Norway |

**Table 3.** $CO_2$ Forcing (W m$^{-2}$) ($G$), Net Feedback (W m$^{-2}$ K$^{-1}$) ($\lambda$), Climate Response (W m$^{-2}$) ($\Delta$CRE), and Equilibrium climate sensitivity (K) (ECS) values.

| Model | $G$ | $\lambda$ | $\Delta$CRE | ECS |
|---|---|---|---|---|
| ACCESS1-0 | 5.78 | -0.74 | 0.11 | 3.90 |
| ACCESS1-3 | 5.71 | -0.80 | 0.27 | 3.57 |
| bcc-csm1-1 | 6.72 | -1.20 | -0.06 | 2.81 |
| BESM-OA2.5 | 8.62 | -1.45 | -0.13 | 2.96 |
| BNU-ESM | 7.45 | -0.92 | -0.27 | 4.04 |
| CanESM2 | 7.51 | -1.02 | 0.16 | 3.67 |
| CCSM4 | 7.27 | -1.23 | -0.15 | 2.96 |
| CNRM-CM5 | 7.34 | -1.12 | -0.19 | 3.28 |
| CSIRO-Mk3-6-0 | 5.01 | -0.60 | 0.25 | 4.21 |
| FGOALS-g2 | 5.59 | -0.83 | -0.08 | 3.36 |
| FGOALS-s2 | 7.58 | -0.90 | -0.45 | 4.20 |
| GFDL-CM3 | 5.91 | -0.74 | 0.49 | 4.00 |
| GFDL-ESM2G | 5.98 | -1.23 | -0.21 | 2.43 |
| GFDL-ESM2M | 6.69 | -1.37 | -0.31 | 2.44 |
| GISS-E2-H | 7.74 | -1.64 | -0.50 | 2.37 |
| GISS-E2-R | 7.26 | -1.69 | -0.46 | 2.15 |
| HadGEM2-ES | 5.77 | -0.62 | 0.37 | 4.63 |
| inmcm4 | 5.74 | -1.38 | -0.10 | 2.07 |
| IPSL-CM5A-LR | 6.38 | -0.79 | 0.70 | 4.04 |
| IPSL-CM5B-LR | 5.25 | -1.00 | 0.29 | 2.63 |
| MIROC5 | 8.95 | -1.66 | -0.43 | 2.69 |
| MIROC-ESM | 8.33 | -0.88 | 0.14 | 4.74 |
| MPI-ESM-LR | 8.07 | -1.10 | -0.06 | 3.67 |
| MPI-ESM-P | 8.39 | -1.20 | -0.04 | 3.49 |
| MRI-CGCM3 | 6.50 | -1.25 | -0.05 | 2.60 |
| NorESM1-M | 6.19 | -1.10 | -0.08 | 2.80 |
| Mean | 6.84±1.09 | -1.09±0.31 | -0.03±0.30 | 3.30±0.76 |