# Peer review of "Assessing the performance of climate change simulation results from BESM-OA2.5 in comparison to a CMIP5 model ensemble"

_Geoscientific Model Development, 2018_

## Referee Comment (RC1) · Anonymous Referee #1 · 4 Jan 2019

In this study, the authors document the climate sensitivity and feedbacks of the Brazilian Earth System Model, ocean-atmosphere coupled version 2.5 (BESM-OA2.5) and compare those characteristics to the CMIP5 ensemble.

There are really two papers co-existing in this study: one focuses on BESM, the other on the CMIP5 ensemble. The first paper appears underdeveloped and is fairly diagnostic, so needs to be deepened. The second paper is essentially an incremental extension of the Andrews et al. (2012) and Vial et al. (2013) studies. The results are interesting but seem out of scope in a GMD paper that really ought to focus on BESM. For these reasons, I recommend major revisions.

The main changes I would like to see are:

- The title is too vague. The paper is not about BESM-simulated climate change in general – that would imply showing results from historical or projection simulations. The paper is in fact about BESM simulated climate sensitivity and feedbacks.

- A re-organisation of Section 2 Model Description. At the moment, it has only one subsection, which is a mixture of model description and comparison to the previous version. This should be split cleanly into two subsections focused on each aspect. The model description should be more complete (i.e. in addition to the aspects listed in Table 1, it should briefly refer to the other elements of the model: Boundary layer, aerosols, convection, dynamical core, gravity waves, large-scale clouds and precipitation)

- The paper spends too long discussing CMIP5 models when it really should be discussing BESM. Three changes would fix the balance. First, Section 3.2 needs to be shortened because it is essentially a re-telling of Andrews et al. (2012) and Vial et al. (2013). In the context of the paper the reader is only interested in the physical meaning of the different variables estimated by the Gregory and kernel methods. Second, the results presented in Sections 4.1 and 4.2 need to be compared to the original papers: are the results replicated? How many models have been added/removed compared to the original papers? Third, a lot of the analysis in Section 4.3 is about CMIP5 models in general (page 10 especially), and that has been said already in other papers so could just be repeated briefly. Instead, the space could be used to deepen the analysis of the BESM simulations, as indicated in my next point.

- The authors frequently compare BESM to the CMIP5 average, or say that it is within the CMIP5 range (which is often large), or note where BESM is an outlier. But such statements are only mildly useful. After all, it may not be a good thing to be close to the CMIP5 average. Instead, readers need evidence for a

deep understanding of why BESM behaves like it does. Why is there a radiative imbalance of 2 W m$^{-2}$? That is a large value. Does that cause a model drift? Does the model conserve energy? Then, why is the 2xCO2 radiative forcing at the higher end of the range? Is it an issue for the radiative transfer code? Then, why is BESM an outlier in terms of cloud feedbacks? The reader is told that the answer lies in the high latitudes (Page 8 line 35 – Page 9 line 1), but what are the mechanisms? Change in low-cloud cover? Change in phase from ice to liquid? Finally, regarding the "warming hole" in the North Atlantic, does BESM simulate it for the reasons listed by Drijfhout et al. (2012)? This is not an exhaustive list: I may have missed other responses that need discussing more deeply.

Other comments:

- Page 2 line 3: The main result of the "trapping" of infrared radiation is an increase in ocean heating content, since this is the Earth system component with the largest heat capacity.

- Page 2 line 20: The wet-gets-wetter etc. is probably too simple and more subtle descriptions are now preferred, see for example Marvel and Bonfils, doi:10.1073/pnas.1314382110 (2013).

- Page 3, line 10: Is the model hydrostatic or not?

- Page 3, line 21: What microphysical processes? Clouds?

- Page 3, line 24: The 2m subscript is confusing. Are the authors talking of diagnostic or prognostic variables here?

- Page 4, section 3.1: It would be useful to refer to the CMIP6 DECK here (Eyring et al. doi:10.5194/gmd-9-1937-2016, 2016) since piControl and abrupt4xCO2 are both mandatory simulations within the DECK. Referring to CMIP6 would make the paper more current.

- Page 4, lines 26–31: Need to move the statements on page 5 lines 27–28 and page 6, lines 15–16 here to list the advantages and limitations of both methods in one place.

- Page 5, line 27: Would be useful to refer to Soden et al. doi:10.1126/science.aau1864 (2018) here.

- Page 7, lines 24-25: That statement needs to be clarified and referenced. Perhaps Zelinka et al doi:10.1175/JCLI-D-12-00555.1, 2013?

- Caption of Figure 2: Please make figure captions standalone by defining all acronyms and variables.

- Figure 4: It would be helpful to put a dashed line at lambda = 0 on each panel, to make easier to see where the feedback parameters switch sign.

---

## Referee Comment (RC2) · Anonymous Referee #2 · 22 Jan 2019

Review of "Overview of climate change in the BESM-OA2.5 climate model" by V.B. Capristrano et al.

Overall assessment and recommendation

This paper compares a selected set of results from a reference (PI-Control) and a climate change simulation (abrupt4xCO2) from the newly presented BESM-OA2.5 model with respective results from other models as derived through analysis of the CMIP5 data base. Generally, BESM-OA2.5 appears to perform reasonably in the CMIP5 context, so the paper could provide a good reference for further dedicated research with this model. However, the paper has a number of severe structural deficits that *need to be overcome by a thorough revision*. Beyond, there are quite a number of baffling statements (or unlucky formulations, to put it more mildly) suggesting that, beyond re-structuring, also some kind of re-thinking may be necessary, in order to yield a more insightful presentation of BESM-OA2.5's merits and shortcomings.

General remarks

As already observed by another reviewer, the paper is severely out of balance in that it dwells too much on discussing (and interpreting) CMIP5 model results, while entering too less into the potential origin of BESM-OA2.5 peculiarities. To deepen existing CMIP5 results, in case of need, for an optimal assessment of BESM-OA2.5 is not necessarily beyond the scope of GMD (as suggested by referee #1), but the focus needs to be on the BESM results an their proper appraisal.

In the current text I find the statements in the last paragraph (p. 12, l. 11ff.) rather strange. The main objective of BESM is not supposed to "show climate sensitivity and thermodynamical responses similar to … CMIP5" but rather "to study the climate system [with a model able] to reproduce changes that are physically understood". Besides, that the latter objective should be pursued by any climate model activity, what does this mean for the present paper and its priorities? Is BESM-OA2.5 in fact planned to be applied for dedicated research questions? Is BESM rather intended to be employed with higher, process resolving resolution? Or is it to be developed towards an Earth System Model with high comprehensiveness? Or is it to be optimized as a testbed for different physical (e.g. cloud) parameterizations? The authors were well-advised to decide for their (future) scientific focus first, and then select the proper diagnostics for their CMIP5 comparison accordingly. Else, the reader will remain wondering why some parameter or process is evaluated here, while another one is not. Moreover, the current use and interpretation of diagnostics is made rather schematically, reflecting too little on the origin of specific BESM features.

Finally, I notice that some basic climatological features of BESM-OA2.5 have already been documented in a current GMD manuscript (Veiga et al., GMDD 2018), of which I am not a reviewer. Comments to Veiga et al. look promising, but I strongly recommend to the editors that the present work should only be accepted together with its companion paper.

Other major issues

The authors use (or rather combine) two ways of calculating radiative feedbacks, viz. the regression method from Gregory et al. (2004) and the individual feedback calculation method from radiative kernels. This is quite recommendable, on principle. However, the methods are not equivalent as the phrase "seemingly redundant" (p. 4; l. 27) is suggesting. The finer points of the methodical difference are not addressed properly in the paper. The kernel methods includes, if no specific measures are taken (as in Vial et al., 2013, or in Chung and Soden, 2015), rapid adjustments directly induced from the $CO_2$ forcing.

More severe, the regression method implies that the radiative feedbacks are consistent with the actual radiative transfer module used in the climate model, while this is not true for the kernel method, if another than the radiative kernel from the actual climate model is used (as is the case here). The authors are apparently aware of this fact (p. 5, l. 25, p. 7, l. 16), but repeatedly fail to appreciate it when interpreting results.

In the same context the authors might also consider to refer to Forster et al. (2016) and Smith et al. (2018) here (beyond Vial et al., 2013), with respect to the options of calculating and interpreting effective radiative forcings, radiative adjustments and feedbacks, and climate sensitivity parameters. Are there abrupt4xCO2 simulations with fixed SST from BESM-OA2.5 that could be included in the discussion? Or are those intended to be analyzed in further BESM studies?

Further, I have some concerns about deriving the ECS (which is for 2xCO2) from 4xCO2 simulations by using a factor 2 (p. 6, l. 22). Is this really a standard method? Then it's certainly at odds with available knowledge (e.g., Boer et al., 2003; Knutti and Rugenstein, 2015). However, the authors could argue that they used the same approximation, crude or not, for all evaluated models.

Even if the focus of the paper were re-directed towards the BESM performance, I still suggest a modified title, for example: "Assessing the performance of climate change simulation results from BESM-OA2.5 in comparison to a CMIP5 model ensemble".

Specific and Technical Remarks

p. 1, l. 8 (Abstract): For the following two sentences I would rather expect a general assessment of BESM rather than pure repetition of specific parameter results. While it is obviously true (and worth mentioning) that BESM-OA2.5 is not an outlier off the CMIP ensemble, its appraisal ought to be more process-directed.

p. 2, l. 1: "…, commonly referred to as", I think this is rather a simplification for less developed models, so "…, sometimes given as" may be preferable.

p. 2, l. 20: There is a formal contradiction here: "… is robust from … models" does not fit with "… uncertainty is likely to arise from … inter-model spread", please reformulate.

p. 3, l. 3: "Differences …", this sentence may be omitted as it is essentially repeated at p. 3, l. 3.

p. 3 l. 16: "… uses BAM … with simpler and computationally cheaper parameterizations"; Does this mean that BESM-OA2.3 uses the original BAM? Why has this been changed and could there be consequences of the simplification for the response behavior of the model as addressed in the present paper?

p. 3, l. 20: From the preceding text, it is puzzling that the simplified model should have a better representation of the ToA radiative budget. I assume, however, that this is a result of more careful parameter tuning (but this is not mentioned). Like referee#1, I also wonder whether this relatively large ToA radiative balance bias leads to a considerable present-day surface temperature bias. Does the coupled atmosphere-ocean model use a flux correction?

p. 3, l. 22: "surface layer"; I assume you mean the "planetary boundary layer", don't you? Or does tis refer to pure diagnostics, as suggested by the following sentence.

p. 4, l. 4: "general mean present-day climate state"

p. 4, l. 7: "BESM-OA2.5 also is capable …"; this sentence is rather vague, are you talking about ocean variability here? Or does this include the leading modes of long-term atmospheric variability like NAO, PNA etc. ?

p. 4, l. 11: "overturning"

p. 4, l. 12: "slightly"

p. 4, l. 14: You might wish to address the matter of storm track variability here, but only if this is supposed to be a field of BESM application in the future. And if it has been actually studied, of course.

p. 4, l. 30: "… the Gregory et al. (2004) method …"; from various reasons it may be preferable to introduce (and refer to) the respective method as "… the regression method …". Mainly, because using the terms "regression" and "radiative kernel" directly points to the methodical differences.

p. 5, l. 17: "… we extract the clear sky radiative flux components from the BESM and CMIP data bases in order to …"

p. 5, l. 25: (see major remarks above) – as the assumption is not necessarily true a remark should be made on the consequence for interpretation in case that there are substantial differences between the radiation modules.

p. 6, l. 4 (and l. 11): No information is given on how stratospheric temperature (and water vapour) changes are accounted for when calculating the feedback

parameters. I recommend at least making a statement, if those contributions are included in the Planck feedback, or if they are shifted to the residuum *Re* (which I guess is, what you actually did). See also Rieger et al. (2017, their Fig. 5).

p. 6, l. 17: "… cloud feedback is approximated using …"; I'm aware that this is a standard method, so the authors are not responsible for the quality of this approximation.

p. 6, l. 22: I expect that the respective 30 year periods are not fully stationary as the deep ocean components of the various models have not reached equilibrium. If your analysis allows, please give some information on the remaining trend in the evaluated periods. Or have the data been de-trended before using them as an input to the radiative kernels?

p. 7, l. 5: "… the spatial inner product …"; the authors might like to introduce the term in this way, but I assume they compute what is elsewhere called the 'Pearson correlation coefficient', hence I recommend to use the latter term through the rest of the paper.

p. 7, l. 9: "These linear regressions …"; this sentence is hard to read and needs rewriting. With the current formulation, it is not possible to unravel for which purpose all-sky or clear-sky data haven been used.

p. 7, l. 11: The values given are at odds with what is written p. 5, l. 12, concerning $G$, $\lambda$, and *ECS*. Please, give an explanation (which is probably to be found in the fact that no actual equilibrium has actually been reached).

p. 7, l. 13: "… similar to those of Andrews et al. …"; in fact, the reader certainly expects no less than this, as those authors used CMIP5 data as well. Where does the difference come from? Interpolation as mentioned on p. 4, l. 23?

In the simulations with BESM, has there any form of "radiation double calling" been used to calculate radiative forcings or feedbacks? That could help to assess whether the radiation parameterization within BESM produces results (largely) consistent with the GFDL and NCAR kernels.

p. 7, l. 28: "Both radiative kernels are used …"

p. 8, l. 4: The following discussion (of Figure 4) is an example in a text flow that is largely out of scope with the paper's focus. Most of this is established knowledge from a multitude of previous papers. A clear change of perspective towards the specific features of BESM is advisable.

p. 8, l. 6: "The faster increase …", I assume you mean "stronger", don't you?

p. 8, l. 7: The two sentences discussing the possible cause-and-effect relation of water vapor and lapse-rate feedbacks is somewhat confusing. The general notion, I think, is the different degree of turbulent mixing in tropical, mid and polar latitudes. I recommend referring to, e.g., Po-Chedley et al. (2018), who draw a lucid and consistent picture of the latitudinal differences.

p. 8, l. 16: "… as noted in yellow and blue shaded areas in Figure 4", this hint would better be given when the discussion of Figure 4 starts (l. 4) or, alternatively" in the figure caption.

p. 8, l. 22: This paragraph is either too short (different cloud feedback results from different methods being a highly complex issue) or too long (as these general issues are not necessarily within the scope of the paper). Please focus on what could be a reason for the specific behavior of BESM in this particular case.

p. 8, l. 35: "This is due to …", a rather technical reasoning (which continues throughout this paragraph). The reader would rather be interested in the physical reason. I the cloud cover response over sea (60°S) and over sea ice (Arctic) less well simulated by BESM compared to land areas? Or could it by that there is a problem with the cloud phase feedback (e.g., Mitchell et al., 1989; Tan et al., 2016) in BESM? I would find it sufficient, if some ideas could be formulated, with hints to future research.

p. 8, l. 6: "stratocumulus region", this is presumably a different entity and not connected to the BESM peculiarities showing up in Figure 4.

p. 9, l. 10: The section 4.3 with its figures 6 and 7 (the scatter plots) is not very insightful to me. What are these correlation diagrams (especially Figure 6) supposed to teach the reader? Is this a standard diagnostic? Does the placement of BESM in the third quadrant reveal anything about this model in a physical sense? Please, give some reasoning for the figure's usefulness in the present paper. Interpretation of precipitation change patterns is more lucid; yet, it would be fine to know whether, e.g., the southward shift of the SPCZ in BESM does occur in other CMIP models too (even if not in the ensemble mean).

p. 9, l. 27: "… near the equator compared to the subtropics …"; ("as opposed to" suggests that the subtropics grow colder)

The statement beginning on p. 10, l. 21 "This increase …" sounds somewhat counter-intuitive and is, in my opinion, an oversimplification of what the cited papers actually say. Rather, the non-linear increase of water vapor available for condensation, as suggested by the Clausius-Clapeyron relation, is limited towards a more linear relation by tropospheric radiative cooling (Mitchell et al., 1987).

p. 10, l. 25: "ACCESS1-0 and HadGEM2-ES use …" up to the end of this paragraph: that may all be true, but the reader would rather be interested whether this implies anything for BESM.

p. 10, l. 32: "… (SLP) response pattern …"

p. 10, l. 30: This whole paragraph gives a lot of (by no means unfounded!) physical reasoning on tropospheric variability patterns, but in the end takes a simple similarity of the SLP mean response patterns from BESM and from the CMIP ensemble to indicate that BESM may well represent such variability patterns. This is a bold conclusion, which in my view would need backing from actual variability pattern analysis. Is such analysis planned?

p. 11, l. 23: "It is shown …", this a very odd 'conclusion', as this statement is common knowledge motivating any research on global warming, and it is certainly not "… shown in this study". Even "… confirmed by this study" would be a summary much too weak for motivating publication of this paper. Please, find a more specific main conclusion that is directed towards the BESM performance.

p. 11, l. 31: Here, some information about the BESM radiation module and its evaluation would emphasize that the radiative feedbacks calculated from BESM output within the CMIP range indeed indicate a good representation of such feedbacks inside that model (see major issues).

p. 12, l. 5: You might delete "However,"; I see no contradiction of this sentence with the preceding one.

p. 12, l. 12: "… is not the aim for the BESM development", this whole paragraph is a very puzzling wrap-up of your paper (see general remarks).

Figure 3, Figure 8: Please ensure that this figure will appear larger in the eventual paper, otherwise it will be hard to decipher.

Caption of Figure 6: "Shaded areas"; this return in several other figure captions, too. You mean the *white* areas, don't you?

References (only if not already cited in the paper):

Boer, G, Yu , B., 2003: Climate sensitivity and climate state, Clim. Dyn. 21, 167-176.

Chung E.S., Soden, B., 2015: An assessment of direct radiative forcing, radiative adjustments, and radiative Ffeedbacks in coupled Ocean-Atmosphere models, J. Clim. 28, 4152-4170.

Forster, P.M., et al., 2016: Recommendations for diagnosing effective radiative forcing from climate models for CMIP6, J. Geophys. Res. 121, 12460-12475.

Knutti, R., Rugenstein, M., 2015: Feedbacks, climate sensitivity and the limits of linear models, Philos. Tr. Roy. Soc. A 373, 20150146.

Mitchell, J.F.B., Wilson, C.A., Cunnington, W.M., 1987: On CO2 climate sensitivity and model dependence of results, Q. J. Roy. Metorol. Soc. 113, 293-322.

Mitchell, J.F.B., Senior, C., Ingram, W., 1989: $CO_2$ and climate – a missing feedback, Nature 341, 132-134.

Po-Chedley et al., 2018: Sources of intermodal spread in the lapse-rate and water vapor feedbacks, J. Climate 31, 3187-3206.

Rieger, V., Dietmüller, S., Ponater, M., 2017: Can feedback analysis be used to uncover the physical origin of climate sensitivity and efficacy differences? Clim. Dyn. 49, 2831-2844.

Smith, C.J. et al., 2018: Understanding rapid adjustments to diverse forcing agents, Geoph. Res. Lett. 45, 12023-12031.

Tan, I., Storelvmo, T., Zelinka, M., 2016: Observational constraints on mixed-phase clouds imply higher climate sensitivity, Science 352, 224-227.

---

## Author Comment (AC5) · 6 Jan 2020

**Reply to the Reviewer 1**

**I thank the authors for addressing my comments. The paper has been improved, although language editing is necessary. The remaining surprising result is that the coupled model, which suffers from a 4 W m-2 radiative imbalance, manages to produce a non-drifting preindustrial control simulation. It seems likely that the coupled model does not conserve energy or that some energy flows are not diagnosed properly. I suggest to mention in the conclusion these imbalance and the potential issues that they hint at.**

[Figure]

We thanks all the reviewer suggestions. An English proofreading was hired. The requested discussion about the radiative imbalance at the top-of-atmosphere is added in the Conclusion section.

---

## Author Comment (AC6) · 6 Jan 2020

**Reply to the Reviewer 2**

**Review of "Assessing the performance of climate change simulation results from BESM-OA2.5 in comparison to a CMIP5 model ensemble" by V.B. Capistrano et al. (revised manuscript)**

**Overall assessment and recommendation**

I regret to conclude that this paper has not been sufficiently improved by the revision process to be acceptable. While I appreciate that the authors have tried to bring the characteristics of BESM-OA2.5 more in focus of their presentation (rather than discussing the general performance of CMIP5 models), the results is still a clumsy and partly dis-organized concatenation of results and result comparisons that do not lead to a clear assessment of the suitability of BESM for specific purposes. New text often has been insufficiently harmonized with the previous text, making reading through the manuscript still an extremely arduous task.

As I stated in my original review, my impression is that BESM is a reasonable model that could be useful for specific applications at least. Hence, I am reluctant to reject this paper once and for all. The authors should be allowed to make one more attempt to create a straightforward paper with a coherent message. To this end (as I have proposed before) the focus of future use of BESM should be made clear, considering the merits and shortcomings of this model. The authors should intensify their attempts to interlink the evidence arising from individual parameter evaluation. This already has been tried in a number of cases, but it too often results in circular reasoning, not approaching the roots of characteristic BESM features. Finally, I emphasize that just executing through my list of technical and language suggestions alone will not do! The author team apparently does not include an English native speaker, hence assistance in producing a proper English text ought to be given by either the editorial office or from some other consultant. Otherwise, I fear that I will be reluctant to read through this paper once again.

Reply: We thank all the reviewer suggestions. The manuscript was rewrite to make each the paragraph message clearer. Furthermore, as requested, an English proof-reading was hired (editing certificate is attached here).

**General remarks**

**1) Section 2.1 still contains elements of a comparison between BESM-OA2.5 and BESM-OA2.3 (e.g., p. 3, l.32) though a dedicated section (2.2) is supposed to cover such differences.**

Reply: This comparison was removed and a new discussion was added in the section 2.2.

**2) It is on occasions still hard to reproduce what has actually been done and why (e.g., p. 6, l.25).**

Reply: Please see the section about Language and Technical Remarks below.

**3) No reason is given on p. 8, 2nd paragraph, why only 11 rather than 15 CMIP5 models are included here. Or are sometimes 11, sometime 15, models used, as could by read out of p. 8, l. 6?**

Reply: Andrews et al. (2012) used 15 CMIP5 models and we used 26, which means that we added 11 models. We reorganized the paragraph.

**4) Occasionally, I still miss a comment on the specific performance of BESM, even if it's well consistent with the CMIP5 ensemble (e.g. Figure 3).**

Reply: The requested information were added.

**5) Page 9, 1st paragraph: This has been reformulated, but is now even more confusing than before. Please reconsider, what is the intended message here, with focus on BESM. Then stick to specific reasoning to underpin that message.**

Reply: The paragraph was rewrite to make the main message clear.

**6) Page 9, 2nd paragraph: Here, too, the line of reasoning remains badly organized: What is the message: Does BESM simulate a stronger Arctic amplification than the CMIP ensemble (suggested by the more negative Planck feedback)? This could simply explain more snow/ice melting. Evidently the lapse rate feedback in BESM is exceptionally positive at Arctic latitudes, pointing at a enhanced vertical gradient in the temperature response. Can this be discussed in the context of the Veiga et al. paper (atmospheric temperature response)?**

Reply: The paragraph was rewrite to make the main message clear.

**7) The last paragraph of section 4.2, with much newly introduced text, is very hard to understand both concerning the weak use of English language and a confusing inherent logic. I have read through this paragraph three times, but then gave up, being unable to reconcile the statements in the text with what the**

**figures display.**

Reply: The paragraph was rewrite to make the main message clear.

**8) Scatter Diagrams in Figures 8 and 9: Do you conclude anything from the apparent correlation between precipitation in piControl and abrupt4xCO2 on one side, and missing correlation for respective surface temperature levels on the other side? Does this have implications for the BESM model performance.**

Reply: BESM results were included and linked previews discussions.

**9) In the last paragraph of the conclusions an outlook to what is planned with BESM-OA2.5 (future research focus) is still lacking. However, this would be the logical outcome of the assessment of its merits and shortcomings, which I assume is what the present paper has been written for.**

**Language and Technical Remarks**

**p. 1, l. 7 (Abstract): " ... the CMIP5 ensemble mean value .. "**

Reply: The climate feedback responses were estimated for 25 CMIP5 models individually and for BESM, no just for the CMIP5 ensemble. This strategy as adopted in order to visualize where BESM in comparison to a distribution of climate response.

**p. 1, l. 8 (Abstract):** " ... BESM simulation show zonally average feedbacks, estimated from radiative kernels, that lie within the ensemble standard deviation ..."

Reply: Done.

**p. 1, l. 11 (Abstract):** "... BESM also features a strong positive ..."

Reply: Done.

**p. 1, l. 12 (Abstract):** As this sentence mentions a merit of BESM, while the preceding sentence comments on a disagreement with CMIP, "moreover" makes quite an unlucky connection. By the way, "consistent" with what?

Reply: "Moreover" was changed to "However". The BESM results are consistent with the CMIP5 ensemble mean. Changes were done to clarify this point.

**p. 2, l. 7:** "... results in a temperature rise ..."

Reply: Done.

**p. 3, l. 7:** "... models, also discussing peculiarities in the BESM climate response."

Reply: Done.

**p. 3, l. 16: "... same as used by Veiga ..."**

Reply: Done.

**p. 3, l. 17: "... model, with its dynamical core being based on ..."**

Reply: Done.

**p. 3, l. 22: "... of physical parameterizations between BAM (as used in this paper) and BAM NWP ..."**

Reply: Done.

**p. 3, l. 24: "... 28 layers, unevenly spaced, in the ..."**

Reply: Done.

**p. 3, l. 29: "... is able to capture ..."**

Reply: Done.

**p. 3, l. 30: "... with a double ITCZ ..."**

Reply: Done.

**p. 3, l. 31: "improvement", despite of the "substantial biases" addressed in the preceding sentence?**

Reply: This sentence was adapted to "Comparison to previous version" section as requested in the general considerations.

**p. 3, l. 33: "... decadal climate variability patterns." This is meant, isn't it?**

Reply: Yes. It is correct now.

**p. 4, l. 4: I understand that AMOC is a circulation structure rather than a parameter. So, what "value" a you referring to? If required, please give an absolute or relative difference of the parameter you have in mind.**

Reply: The AMOC strength simulated by the model in the piControl is around 14 Sv (for 1000 years). The AMOC strength observed by the RAPID project is roughly 17 Sv (McCarthy et al. 2015).

[Figure]

(see the supplementary material) Fig1. - Maximum AMOC simulated by the piControl from the beginning of the simulation up to 1000 years. The red dash-dot lines shows the linear trend.

**p. 4, l. 9: "... are determined, which are important ...". Anyway, the content of this sentence to me resembles what is given below (p. 4, l. 14), with the sentences in between (starting with "The total energy balance ...") causing an awkward logical break.**

Reply: In order to avoid this apparent logical break, the sentence "which are important in the coupling between atmosphere and ocean" was removed. The section emphasizes the main differences between BESM versions, that are in the atmospheric model parametrization, specially in the way the diagnostic surface layer variables are calculated. The general differences in the atmospheric model are discussed first, and just after this is introduced more details about surface layer variables (where was found a repetition about the importance of these variables for the ocean-atmosphere coupling).

**p. 4, l 19: This sentence again repeats what is given in p. 4, l. 9 ...**

Reply: Please, see the immediately above answer.

**p. 5, l. 6: "... which means a spin-up of 150 years." Does this mean that the 150 yrs of abrupt4xCO2 are regarded as a spin-up here (due to their non-equilibrium character)? Or are 150 yrs of abrupt4xCO2 swapped as a spin-up, and another**

**150 yrs evaluated as some kind of quasi-equilibrium? Please, clarify.**

Reply: The piControl spin-up of 150 years means that the piControl run for 150 years before the analysed period. Therefore, after the 150 yr run, two new simulations of 150 yr are started: 1) the piControl continuation run; 2) the Abrupt4XCO2 run. New informations are added to manuscript text.

**p. 5, l. 6: "... commonly employed ... for climate change assessment"; please, be careful to distinguish between "climate change assessment" and "climate sensitivity assessment"! In my view, "climate change" in the CMIP context is rather assessed through historical simulations and future scenario simulations.**

Reply: "...climate change assessment" is changed to "...climate sensitivity assessment" in the new version.

**p. 5, l. 12: In this paragraph the "forth and back" jumping in addressing the merits of the regression and kernel method is somewhat confusing but could be easily avoided.**

Reply: The "forth and back" jumping was avoided in this version.

**p. 5, l 27, 28: There's still something wrong with the sentences here. Suggestion: "As G can be approximated by backward regression towards $\triangle$Tas=0, ECS can be estimated as ECS$=-$G/$\lambda$."**

Reply: The alteration proposed was done. The intention with the original sentence was emphasize the computation economy that the regression method allows, avoiding a simulation in a order of millenia. The following sentence was removed: "For this method the ECS can be estimated as ECS=-G/$\lambda$ in a shorter simulation (typically of 150 year) without reach the thermodynamical equilibrium.".

**p. 5, l 30: "... it is common to divide the result derived from 4xCO2 simulations by 2 (Andrews ..."**

Reply: Done.

**p. 6, l. 9: "... is used next, in order to partition the ...", as "next to" is confusing. By the way, "separate" or "split" may be preferred to "partition".**

Reply: Done. It was used "decompose" instead of "partition"

**p. 6, l. 14: "integraly" -> "fully" (or "necessarily")**

Reply: Done. "integrally" was replaced by "necessarily" .

**p. 6, l. 16: "This, however, assumes that ..."**

Reply: Done.

**p. 6, equation 3: Tas is the near surface temperature (p. 5), but what is then Ts? I tried to clarify this by looking into Vial et al. (2013), without success. Please, be precise in citing, or explaining what you have done, and why.**

Reply: Ts means surface temperature whereas Tas means near-surface atmospheric temperature. In order to avoid misunderstandings, besides Vial et al. (2013) was cited Soden and Held (2006, page 3356), which has a good compatibility with the variables presented in the equation 3 of our work.

**p. 6, l. 25: Confusing: As q is in the data base, why should it be approximated based on the assumption of constant relative humidity? To my knowledge, this is not common in feedback analysis. Is it possible that you are misinterpreting the cited references here?**

Reply: The assumption of constant relative humidity is associated with how the water vapor kernel is obtained. For water vapor kernel, it is computed the specific humidity change corresponding to a 1-K increase (holding relative humidity constant). Please see Soden et al. (2008) page 3509 and Shell et al. (2008) page 2271. Additional information about the necessity of this assumption was included in consonance with what was requested in the general comments.

**p. 7, l. 5: "... changes are not accounted ..."**

Reply: Done.

**p. 7, equation 5: It is not immediately obvious, what the indices "a" and "k" mean. The index "k" means the change in △CRE due to the noncloud feedbacks, while the index "a" means the △CRE adjusted (to obtain the cloud feedback).**

Reply: Additional information was provided to this new manuscript version.

**p. 8, l. 4: "... were assessed as it was performed ..."; I do not understand this sentence. Are the data not from the ESGF data base (p. 5, l. 10) ??**

Reply: The sentence is to inform the reader that we used the same method (analysis, assessment or evaluation) realized by Andrews et al. (2012). This is not supposed to be link with a mention to ESGF.

**p. 8, l. 6: "... e inmcm4 ", did you intend "... and inmcm4"?**

Reply: Done.

**p. 8, l. 13: witch -> which**

Reply: Done.

**p. 8, l. 29: Please, explain how Figure 4 is related to Figure 3. Is it simply an average over the latitudinal profile of Fig. 3? Your discussion of the Planck feedback is casting doubts concerning this: If it's constant by about -4 Wm-2K-1 (l.29) with mostly negative deviations at polar latitudes, how can this result in a global mean of -3.6 Wm-2K-1 ( l. 26)? Please, cross-check the numbers.**

Reply: a) The Figure 3 shows the global mean for the climate feedbacks, where is possible note the models dispersion. The Figure 4 shows the same feedbacks (and the Planck feedback) but for the zonal average.

b) The ensemble Planck feedback is about -3.39 W/m-2K-1 at the Equator (as well as in the Tropics). It has values below -10 near North Pole, however, we can not forget that the global mean is calculated considering the areal weight for each latitude, which is smaller for polar zones. Therefore, the global mean features a value around -3.6 W/m-2K-1.

**p. 8, l. 32: "stronger vertically homogeneous warming". This is a strange reasoning, as the Planck feedback is essentially the surface warming, constantly extrapolated upward through the depth of the troposphere. Can the message of this sentence be reconciled with Figure 8?**

Reply: a) We totally agree with the comments. By definition the Planck feedback assumes that the temperature change is vertically uniform throughout the troposphere with respect to surface (Soden et al. ,2008, page 3515). This is in the Eq. (4) of the manuscript:

This also is in accordance with what is stated by Jonko et al. (2013): "The Planck feedback is the response of longwave (LW) TOA flux to a perturbation in

surface temperature that is applied to each vertical layer of the troposphere". On the other hand, the lapse-rate feedback is related to the radiative response to changing the vertical temperature structure. Therefore, it was added more information regarding the relation of the Planck feedback and surface temperature.

b) It add more information mentioned the link between Figure 8 and results from figures 3 and 4.

**p. 9, l. 20: The partly revised text in this paragraph (see also major comments) contains some sensible elements, but is also moving in circles, explaining stronger sea-ice melting with stronger surface warming and vice versa. More re-organisation of the text is necessary.**

Reply: Modifications were performed as requested in the major remarks.

**p. 9, l. 22: "Those negative values ...", it is unclear which values are addressed.**

Reply: It is about negative Planck feedback. Such paragraph was reorganized.

**p. 9, l. 30: "The highest positive values ...", I would expect that backscattering increases if ice turns into water, driving the shortwave cloud feedback to more negative values. However, your later discussion (Figure 7, see also below) seems to suggest that the longwave cloud feedback is the dominant component.**

Reply: This whole paragraph was rewrite. Since the ice has a greater albedo

than water, when occurs sea-ice melting the albedo decreases, consequently, the outgoing shortwave radiation at the TOA also decreases. Two aspects are highlighted in the high latitudes for BESM cloud feedback: a weak increase in total cloud cover, which contributes to a negative SW cloud feedback (Figure 6a-b); and a low-level clouds upward shifting that is responsible for a gain of LW energy, which is related to sea-ice melting and indirect linked to albedo feedback cloud mask (Figure 6 c-d).

**p. 9, l. 31: I feel that the following text (until "... outlier for the cloud feedbacks.") is mainly repetitive.**

Reply: I was rewrite.

**p. 10, l. 4: "$\lambda$a, $\lambda$ac", are you referring to an analytical framework that is given in Cess et al. (1989)? Otherwise the reader is rather left in the dark here.**

Reply: They are in the Equation (5). "$\lambda$a, $\lambda$ac" are the albedo feedback and the albedo feedback for clear-sky, respectively. More information is added to clarify the discussion.

**p. 11, l. 4: "models with ... apparently do not show ..."; please also replace "present" by "show" on many occasions thereafter.**

Reply: Done.

**p. 11, l. 24: "...quadrupling of atmospheric CO2 with the piControl pre-industrial CO2 concentrations ...": meaning what? The two first sentences of this paragraph appear to transport the same statement.**

Reply: Real meaning is: "..quadrupling of atmospheric CO2 with respect to the piControl pre-industrial CO2 concentrations . . .". It was changed for the new version.

**p. 11, l. 28: "... precipitation increase is not governed ..."**

Reply: Done.

**p. 11, l. 31: Does this have in any way implications for the use of these somewhat "outlying" models?**

Reply: The fact that a model is an outlier in one feature does not invalidate that model in others features. For example, HadGEM2 is widely recognized for having a good representation of precipitation in many parts of the globe; however, it is on the list indicated in the manuscript that models do not have a linear fit between global warming and precipitation change. Such behaviour may be due to chosen tuning in physical parameterization.

**p. 12, l. 13: "...regions with the strongest increase of westerly winds at all levels indicate a southward jet displacement ..."**

Reply: Done.

**p. 12, l.18: Is "omega" something different from "vertical velocity"? Anyway, "omega" isn't self-explaining, so please adjust the text.**

Reply: Omega is related to vertical velocity, but is not the same variable. Omega is Dp/Dt (isobaric coordinates), while vertical velocity is w=Dz/Dt (height coordinates). For hydrostatic approximation Dp/Dz = - â■ť'g with â■ť' constant, Omega = -â■ť'g w. In order to clarify the sentence we changed "vertical velocity" to "omega vertical motion".

**p. 12, l. 31: "... radiative code transference ...", do you mean "performance"? Is there any indication of that particular feature for BESM's radiative transfer model?**

Reply: It is related to BESM's radiative transfer model. The correction was done.

**p. 12, l. 31: "... rapid adjustments ..."; the rapid adjustment process is included in the CMIP5 model results as well, per construction. You apparently did not calculate the rapid adjustments for BESM, but do you have any indications that there might be a systematic bias with respect to CMIP (see Smith et al., 2018).**

Reply: We did not integrated the BESM (atmosphere-only: BAM) model with climatological SST and ice cover doubling CO2 in order to evaluate the rapid adjustments. However, we think that this could be done a future study.

**p. 13, l. 4: "Two regions indicate enhanced inter-model standard deviation for Planck, lapse-rate and albedo feedback"; also in the rest of this paragraph the use of English language is very weak, making the meaning nearly incomprehensible for me.**

Reply: The entire paragraph has been rewritten and a third party English proof-reading service has been performed.

**References (only if not already cited in the paper):**

Smith, C.J. et al., 2018: Understanding rapid adjustments to diverse forcing agents, Geoph. Res. Lett. 45, 12023-12031.

**References cited in the responses:**

Andrews, T., Gregory, J. M.,Webb, M. J., and Taylor, K. E.: Forcing, feedbacks and climate sensitivity in CMIP5 coupled atmosphere-ocean climate models, Geophysical Research Letters, 39, n/a–n/a, https://doi.org/10.1029/2012GL051607, 2012.

McCarthy, G.D.; Smeed, D.A.; Johns, W.E.; Frajka-Williams, E.; Moat, B.I.; Rayner, Darren.; Baringer, M.O.; Meinen, C.S.; Collins, J.; Bryden, H.L. (2015): Measuring the Atlantic Meridional Overturning Circulation at 26°N, Progress in Oceanography, 130:

91-111. doi:10.1016/j.pocean.2014.10.006

Jonko, A. K., Shell, K. M., Sanderson, B. M., and Danabasoglu, G.: Climate Feedbacks in CCSM3 under Changing CO 2 Forcing. Part II: Variation of Climate Feedbacks and Sensitivity with Forcing, Journal of Climate, 26, 2784–2795, https://doi.org/10.1175/JCLI-D-12- 00479.1, 2013.

Shell, K. M., Kiehl, J. T., and Shields, C. a.: Using the radiative kernel technique to calculate climate feedbacks in NCAR's Community Atmospheric Model, Journal of Climate, 21, 2269–2282, https://doi.org/10.1175/2007JCLI2044.1, 2008.

Soden, B. and Held, I.: An Assessment of Climate Feedbacks in Coupled Ocean – Atmosphere Models, Journal of Climate, 19, 3354–3360, https://doi.org/10.1175/JCLI9028.1, 2006.

Soden, B. J., Held, I. M., Colman, R., Shell, K. M., Kiehl, J. T., and Shields, C. A.: Quantifying Climate Feedbacks Using Radiative Kernels, Journal of Climate, 21, 3504–3520, https://doi.org/10.1175/2007JCLI2110.1, 2008.
* * *
[Figure]

[Figure]

**Fig. 1.** Maximum AMOC simulated by the piControl from the beginning of the simulation up to 1000 years. The red dash-dot lines shows the linear trend.

---

## Author Response (AR1)

**Reply to the Reviewer 1 (updated with corrected indication of changes)**

First of all, we would like to thanks the extraordinary review. It is evident the importance of your suggestions, which is associated with the quality and relevance of all information for GDM reader. The original manuscript was planned to intercompare BESM climate model with CMIP5 ensemble, documenting the well-known physical responses to increased CO2. Therefore, many analysis (tables and figures) were proposed with this view, having side-by-side BESM and CMIP5. We agree with the main issue pointed out by both reviewers, that the GMD reader would not be interested if BESM has cli-

mate sensitivity within ensemble dispersion. Thinking in this way, we rewrite parts of the manuscript (following the reviewers' suggestions) where comparisons BESM vs. CMIP5 were mentioned, bringing more discussion about BESM response. Moreover, new figures focusing on BESM results was added, however the original figures and tables remained without change.

1. **The title is too vague. The paper is not about BESM-simulated climate change in general – that would imply showing results from historical or projection simulations. The paper is in fact about BESM simulated climate sensitivity and feedbacks.**
   Reply: According to the suggestion of the anonymous Reviewer 2, the article title was changed to "Assessing the performance of climate change simulation results from BESM-OA2.5 in comparison to a CMIP5 model ensemble".

2. **A re-organisation of Section 2 Model Description. At the moment, it has only one subsection, which is a mixture of model description and comparison to the previous version. This should be split cleanly into two subsections focused on each aspect. The model description should be more complete.**
   Reply: The Section 2 was split in two parts as requested. Moreover, the model configuration was more detailed. Please, see page 3 lines 11.

3. **The paper spends too long discussing CMIP5 models when it really should be discussing BESM. Three changes would fix the balance**

   (a) **First, Section 3.2 needs to be shortened because it is essentially a re-telling of Andrews et al. (2012) and Vial et al. (2013). In the context of the paper the reader is only interested in the physical meaning of the**

[Figure]

**different variables estimated by the Gregory and kernel methods.**
Reply: As far as we have two different methods, we decided explicit all calculations. It worth noting that the first technique is the same as Andrews et al. (2012), however the other does not share the same methods with Vial et al. (2013). The radiative kernel method applied here is similar to its origin paper (Soden et al., 2008), whereas Vial et al. (2013) separated the feedback and the rapid adjustment using different protocols run (see next question).

(b) **Second, the results presented in Sections 4.1 and 4.2 need to be compared to the original papers: are the results replicated? How many models have been added/removed compared to the original papers?**
Reply: As mentioned by the reviewer, the climate sensitivities of 26 CMIP5 coupled models (including BESM-OA2.5) were assessed using the Gregory et al. (2004) linear regression between net radiation in TOA and surface temperature changes, as well as it was performed by Andrews et al. (2012) for 15 CMIP5 coupled models. In the present work, we included the following models: ACCESS1-0, ACCESS1-3, bcc-csm1-1, BESM-OA2.5, BNU-ESM, CCSM4, FGOALS-g2, FGOALS-s2, GISS-E2-H, GISS-E2-R, e inmcm4. For the 15 same models, we found similar results with respect to Andrews et al. (2012). Such small difference may we can attribute grid interpolation as explained in line 4 of page 8. In order to partitioned the feedback agents we used the radiative kernel described in Soden and Held (2006) and Soden et al ( 2008) and Shell et al (2008). In turn, Vial et al. (2013) adapted this previous methodology to consider the tropospheric adjustment to $CO_2$ (comparison between abrupt4xCO2 and sstClim4xCO2, instead of abrupt4xCO2 and piControl).

(c) **Third, a lot of the analysis in Section 4.3 is about CMIP5 models in general (page 10 especially), and that has been said already in other papers so could just be repeated briefly. Instead, the space could be**

[Figure]

**used to deepen the analysis of the BESM simulations.**
Reply: New information was added to include what was requested.

4. **The authors frequently compare BESM to the CMIP5 average, or say that it is within the CMIP5 range (which is often large), or note where BESM is an outlier. But such statements are only mildly useful. After all, it may not be a good thing to be close to the CMIP5 average. Instead, readers need evidence for a deep understanding of why BESM behaves like it does.**

   (a) **Why is there a radiative imbalance of 2 W m$^{-2}$? That is a large value. Does that cause a model drift? Does the model conserve energy?**
   Reply: The AGCM stand-alone run shows a net radiation at TOA of 0.25 W m$^{-2}$ during 20 years of simulation (Fig. 1a). Such radiative imbalance is within the range simulated by different atmospheric models. However, in the coupled simulation, the net radiation imbalance at TOA is amplified up to -4 W m$^{-2}$ (Fig. 1b). The reason for such imbalance is related to higher loss of energy at TOA both from the outgoing long-wave radiation (OLR) and outgoing short-wave radiation (OSR), compared with AGCM stand-alone simulation (Fig. 1c and 1d). In Fig. 1c and 1d, the solid lines represent the coupled model and the dashed lines represent the AGC. The higher loss of energy through the outgoing short-wave radiation is potentially duo to enhanced cloud formation in the coupled model run.

   (b) **Then, why is the 2xCO2 radiative forcing at the higher end of the range? Is it an issue for the radiative transfer code?**
   Reply: BESM-OA2.5 was integrated with UKMET radiative code for SW and LW in order to compare the imbalance of the first year, which is a proxy to the Instantaneous Radiative Forcing.

   (c) **... why is BESM an outlier in terms of cloud feedbacks? The reader is told that the answer lies in the high latitudes, but what are the mechanisms? Change in low-cloud cover? Change in phase from ice to**

[Figure]

**liquid?**

Reply: It is evident from figures presented in the manuscript, that BESM is an outlier for the cloud feedbacks. This is due to a strong shortwave component response over both the Arctic and the Southern Ocean near Antarctica. Considering the SW CRE/ðİŽěTas [described by Cess et al. (1989)] and the individual components of feedbacks cloud mask, we can note that those higher values cloud feedback are mainly consequences of the sum of SW CRE/$\Delta T_{as}$ and the cloud masking for albedo feedback [-(ðİlJĘa-ðİlJĘac)], as shown in Figure 2. For Arctic region, the major contributor for BESM be an outlier is the SW CRE, while for over the ocean near the Antarctic is the albedo feedback cloud mask. In this latter, since the radiative kernel for both all- and clear-sky are the same throughout the models, the difference among them is due to the albedo change [$\Delta a/\Delta T(K_a - K_a^{cs})$]. Over the both regions (Arctic and near Antarctic), an increase in cloud fraction above 850 hPa and a decrease below that level for BESM is observed, which means a low-level clouds upward shifting . Moreover, the increase in cloud cover above 850 hP is stronger than the reduction below (Figure 3a). As consequence, a negative SW CRE change is present in those regions, that is that response to the increase in sun shading (Figure 3b). However, the SW cooling is smaller than the heating provided by LW radiation, as presented in the net effect (Figure 3d). The net radiation heating change is more intense around 60oS, that can be related to the more intense surface albedo change. We could not investigate the change in phase from ice to liquid because we did not designed the experiments to have the liquid and ice water content in their outputs. We pretend develop a new analysis about it in a next work.

(d) **Finally, regarding the "warming hole" in the North Atlantic, does BESM simulate it for the reasons listed by Drijfhout et al. (2012)?**

Reply: A new work about the "warming hole" is in preparation by Nobre et al (2019), which will have more information about BESM transient responses

to radiative forcing in that region.

5. Other comments:

- **Page 2 line 3: The main result of the "trapping" of infrared radiation
  is an increase in ocean heating content, since this is the Earth system
  component with the largest heat capacity.**
  Reply: Done (p.2 l.6)

- **Page 2 line 20: The wet-gets-wetter etc. is probably too simple and
  more subtle descriptions are now preferred, see for example Marvel
  and Bonfils, doi:10.1073/pnas.1314382110 (2013).**
  Reply: Done (p.2 l.26)

- **Page 3, line 10: Is the model hydrostatic or not?**
  Reply: It is hydrostatic. This is information was add in the Model description
  section (p. 3 l. 17)

- **Page 3, line 21: What microphysical processes? Clouds?**
  Reply: It is about the microphysical parameterization of precipitation.

- **Page 3, line 24: The 2m subscript is confusing. Are the authors talking
  of diagnostic or prognostic variables here?**
  Reply: Those variables are diagnostic for the atmospheric model, however
  it is important in the ocean-atmosphere coupling (p. 4 l.15).

- **Page 4, section 3.1: It would be useful to refer to the CMIP6 DECK
  here (Eyring etal. doi:10.5194/gmd-9-1937-2016, 2016) since piControl
  and abrupt4xCO2 are both mandatory simulations within the DECK.
  Referring to CMIP6 would make the paper more current.**
  Reply: Done (p. 5 l. 7).

- **Page 4, lines 26–31: Need to move the statements on page 5 lines 27–
  28 and page 6, lines 15–16 here to list the advantages and limitations**

[Figure]

**of both methods in one place.**
Reply: As far as we decided maintain a separated description of those methods (as discussed previously), we also let the limitation and advantages in different sections.

- **Page 5, line 27: Would be useful to refer to Soden et al. doi:10.1126/science.aau1864 (2018) here.**
  Reply: Done (p.6, l. 16)

- **Page 7, lines 24-25: That statement needs to be clarified and referenced. Perhaps Zelinka et al doi:10.1175/JCLI-D-12-00555.1, 2013?**
  Reply: New information based on the Methods section was provided.

- **Caption of Figure 2: Please make figure captions standalone by defining all acronyms and variables.**
  Reply: Done.

- **Figure 4: It would be helpful to put a dashed line at lambda = 0 on each panel, tomake easier to see where the feedback parameters switch sign.**
  Reply: Done.

**Complete Figure Captions**
Figure 1 – Net of the radiation of TOA simulated by (a) stand-alone AGCM for 20 yearsand (b) BESM-OA2.5 Historical for the first 20 years (1850-1870). (c) and (d) areoutgoing long-wave radiation and outgoing short-wave radiation, respectively. In (c)and (d) the solid lines represent the coupled model and the dashed lines represent the AGCM. Units are in W m-2.

Figure 2. SW Cloud feedback and the albedo and SW humidity feedbacks cloud mask-ing for the CMIP5 multi-model ensemble-mean (solid line) and BESM-OA2.5 (solid linewith dots). Inter-model standard deviations for each latitude are in yellow.

[Figure]

In blue are the feedback limits based on the maximum and minimum values for each latitudeamong the models, not including BESM-OA2.5.

Figura 3. Vertical profiles of the zonal mean of the 4xCO2 - piControl mean difference-for the following variables: (a) Cloud fraction, Radiative heating-cooling rate (dT/dt) of(b) shortwave, (c) longwave and (d) sum of shortwave and longwave.

———————————————

[Figure]

[Figure]

**Fig. 1.**

[Figure]

[Figure]

**Fig. 2.**

[Figure]

(a)

(b)

(c)

(d)

**Fig. 3.**

[Figure]

Geosci. Model Dev. Discuss.,
https://doi.org/10.5194/gmd-2018-209-AC4, 2019

[Figure]

First of all, we would like to thanks the extraordinary review. It is evident the importance of your suggestions, which is associated with the quality and relevance of all information for GDM reader. The original manuscript was planned to intercompare BESM climate model with CMIP5 ensemble, documenting the well-known physical responses to increased CO2. Therefore, many analysis (tables and figures) were proposed with this view, having side-by-side BESM and CMIP5. We agree with the main issue pointed out by both reviewers, that the GMD reader would not be interested if BESM has cli-

mate sensitivity within ensemble dispersion. Thinking in this way, we rewrite parts of the manuscript (following the reviewers' suggestions) where comparisons BESM vs. CMIP5 were mentioned, bringing more discussion about BESM response. Moreover, new figures focusing on BESM results was added, however the original figures and tables remained without change.

1. **... the paper is severely out of balance in that it dwells too much on discussing (and interpreting) CMIP5 model results, while entering too less into the potential origin of BESM-OA2.5 peculiarities. [...] the focus needs to be on the BESM results and their proper appraisal.**
   Reply: Please see the specific and technical remarks.

2. **In the current text I find the statements in the last paragraph (p. 12, l. 11ff.) rather strange. The main objective of BESM is not supposed to "show climate sensitivity and thermodynamical responses similar to ... CMIP5" but rather "to study the climate system [with a model able] to reproduce changes that are physically understood". Besides, that the latter objective should be pursued by any climate model activity, what does this mean for the present paper and its priorities?**
   Reply: Please see the specific and technical remarks.

3. **The authors use (or rather combine) two ways of calculating radiative feedbacks, viz. the regression method from Gregory et al. (2004) and the individual feedback calculation method from radiative kernels. This is quite recommendable, on principle. However, the methods are not equivalent as the phrase "seemingly redundant" (p. 4; l. 27) is suggesting... The kernel methods includes [...] rapid adjustments directly induced from the CO2 forcing. More severe, the regression method implies that the radiative feedbacks are consistent with the actual radiative transfer module used in the**

[Figure]

**climate model, while this is not true for the kernel method, if another than the radiative kernel from the actual climate model is used (as is the case here). The authors are apparently aware of this fact (p. 5, l. 25, p. 7, l. 16), but repeatedly fail to appreciate it when interpreting results.**
Reply: The manuscript was changed to include this concerns. Please, see answers in the specific remarks section.

4. **In the same context the authors might also consider to refer to Forster et al. (2016) and Smith et al. (2018) here (beyond Vial et al., 2013), with respect to the options of calculating and interpreting effective radiative forcings, radiative adjustments and feedbacks, and climate sensitivity parameters. Are there abrupt4xCO2 simulations with fixed SST from BESM-OA2.5 that could be included in the discussion? Or are those intended to be analyzed in further BESM studies?**
Reply: We have not performed an abrupt4xCO2 with fixed SST. We know that it is important to find the rapid adjustment of the troposphere and surface. Therefore, we intended to analyze this issues in a next BESM version.

5. **Further, I have some concerns about deriving the ECS (which is for 2xCO2) from 4xCO2 simulations by using a factor 2 (p. 6, l. 22). Is this really a standard method? Then it's certainly at odds with available knowledge (e.g., Boer et al., 2003; Knutti and Rugenstein, 2015). However, the authors could argue that they used the same approximation, crude or not, for all evaluated models.**
Reply: We used the same method of Andrews et al (2002), that obtained the ECS (for 2xCO2) from comparison between piControl and Abrupt4xCO2.

6. **Even if the focus of the paper were redirected towards the BESM performance, I still suggest a modified title, for example: "Assessing the performance of climate change simulation results from BESM-OA2.5 in compari-**

[Figure]

**son to a CMIP5 model ensemble".**
Reply: Done

7. **Specific and Technical Remarks**

- **p. 1, l. 8 (Abstract): For the following two sentences I would rather expect a general assessment of BESM rather than pure repetition of specific parameter results. While it is obviously true (and worth mentioning) that BESM-OA2.5 is not an outlier off the CMIP ensemble, its appraisal ought to be more process directed.**
  Reply: The abstract was modified in order to attend what was requested.

- **p. 2, l. 1: "..., commonly referred to as", I think this is rather a simplification for less developed models, so "..., sometimes given as" may be preferable.**
  Reply: Done (p. 2 l. 5)

- **p. 2, l. 20: There is a formal contradiction here: "... is robust from ... models" does not fit with "... uncertainty is likely to arise from ... inter-model spread", please reformulate.**
  Reply: After changes in the paragraph, the sentence became out of context, then was removed.

- **p. 3, l. 3: "Differences ...", this sentence may be omitted as it is essentially repeated at p. 3, l. 3.**
  Reply: Done.

- **p. 3 l. 16: "... uses BAM ... with simpler and computationally cheaper parameterizations"; Does this mean that BESM-OA2.3 uses the original BAM? Why has this been changed and could there be consequences of**

Interactive
comment

[Figure]

**the simplification for the response behavior of the model as addressed in the present paper?**

Reply: As required by the other Referee, more information about physical parameterization was included in the manuscript. Moreover, it was mentioned that BAM is the atmospheric component of the climate model BESM-OA2.5. For the current study we used a different parameterization set from that of the evaluation paper of BAM (Figueroa et al. 2016), mainly because a computationally cheap set is desirable in a long simulation. However, changes in those sets result in different climate change response. For instance, a different radiative scheme probably will lead to a different radiative forcing (p. 3, l. 11).

• **p. 3, l. 20: From the preceding text, it is puzzling that the simplified model should have a better representation of the ToA radiative budget. I assume, however, that this is a result of more careful parameter tuning (but this is not mentioned). Like referee#1, I also wonder whether this relatively large ToA radiative balance bias leads to a considerable present-day surface temperature bias. Does the coupled atmosphere-ocean model use a flux correction?**

Reply: A simulation with the atmospheric component only (BAM) presents a imbalance of 0.25 W m$^{-2}$. The imbalance of -4 W $^{-2}$ is related to higher loss of energy at TOA both from the outgoing long-wave radiation and outgoing short-wave radiation, compared with AGCM stand-alone simulation. Despite of this constant imbalance, the surface temperature is in thermodynamic equilibrium in a piControl run. BESM adopted the coupling strategy of pass variables through the surface interface instead of flux. It means that ocean component receive atmospheric variables and calculate the fluxes from atmosphere to ocean, then return variables for atmospheric component in order to calculate fluxes from ocean to atmosphere. Moreover, we do not apply flux correction for our simulations.

[Figure]

- **p. 3, l. 22: "surface layer"; I assume you mean the "planetary boundary layer", don't you? Or does tis refer to pure diagnostics, as suggested by the following sentence.**
  Reply: The surface layer is the lowest layer of the planetary boundary layer.

- **p. 4, l. 4: "general mean present-day climate state"**
  Reply: Done.

- **p. 4, l. 7: "BESM-OA2.5 also is capable ..."; this sentence is rather vague, are you talking about ocean variability here? Or does this include the leading modes of long-term atmospheric variability like NAO, PNA etc. ?**
  Reply: It is about leading modes of long-term climate variability. The sentence "manly that related to Atlantic Ocean", that can contribute to this misunderstand, was removed.

- **p. 4, l. 11: "overturning"**
  Reply: Done.

- **p. 4, l. 12: "slightly"**
  Reply: Done.

- **p. 4, l. 14: You might wish to address the matter of storm track variability here, but only if this is supposed to be a field of BESM application in the future. And if it has been actually studied, of course.**
  Reply: The Storm Track variability of BESM has not been studied yet. This issue will be investigated in a future work.

- **p. 4, l. 30: "... the Gregory et al. (2004) method ..."; from various reasons it may be preferable to introduce (and refer to) the respective method as "... the regression method ...". Mainly, because using the terms "regression" and "radiative kernel" directly points to the methodical differences.**
  Reply: Done.

[Figure]

- **p. 5, l. 17: "... we extract the clear sky radiative flux components from the BESM and CMIP data bases in order to ..."**
  Reply: Done.

- **p. 5, l. 25: (see major remarks above) – as the assumption is not necessarily true a remark should be made on the consequence for interpretation in case that there are substantial differences between the radiation modules.**
  Reply: As suggested, new information about radiative kernel limitation has been written in the Methods section.

- **p. 6, l. 4 (and l. 11): No information is given on how stratospheric temperature (and water vapour) changes are accounted for when calculating the feedback parameters. I recommend at least making a statement, if those contributions are included in the Planck feedback, or if they are shifted to the residuum Re (which I guess is, what you actually did). See also Rieger et al. (2017, their Fig. 5).**
  Reply: Differently from Rieger et al. (2017), the stratospheric adjustment was not investigated here. All feedback calculation was obtained integrating from the surface up to the tropopause. Thereby, it is mentioned that the stratospheric changes are shifted to the residuum (p. 7, l. 5).

- **p. 6, l. 17: "... cloud feedback is approximated using ..."; I'm aware that this is a standard method, so the authors are not responsible for the quality of this approximation.**
  Reply: Done.

- **p. 6, l. 22: I expect that the respective 30 year periods are not fully stationary as the deep ocean components of the various models have not reached equilibrium. If your analysis allows, please give some information on the remaining trend in the evaluated periods. Or have the data been de-trended before using them as an input to the radiative**

[Figure]

**kernels?**

Reply: The time-scale to a coupled model reach a thermodynamic equilibrium is more than 1000 years. Therefore, it is true that the stationary phase is not reached in the analyzed period. We proceeded similar to Vial et al. (2013), that used a 10-year period centered around the 130th year after the CO2 quadrupling in abrup4xCO2. The models data were not submitted to de-trended in the feedback estimation through kernel method.

- **p. 7, l. 5: "... the spatial inner product ..."; the authors might like to introduce the term in this way, but I assume they compute what is elsewhere called the 'Pearson correlation coefficient', hence I recommend to use the latter term through the rest of the paper.**
  Reply: The "spatial inner product" is similar to the "Pearson correlation coefficient" applied to space instead of time as it is commonly used. Therefore, to distinguish between the application for space and time we used the term "spatial inner product".

- **p. 7, l. 9: "These linear regressions ..."; this sentence is hard to read and needs rewriting. With the current formulation, it is not possible to unravel for which purpose all-sky or clear-sky data haven been used.**
  Reply: The application of all-sky and clear-sky is explained in Method section: "... we decompose the feedback parameter into shortwave (SW) and longwave (LW) radiation components and we extract the clear sky radiative flux components from the BESM and CMIP data bases in order to estimate the cloud radiative forcing or cloud radiative effect CRE defined as the difference between the all-sky and clear-sky feedback parameters Andrews et al. (2012)" (p.6 l. 4).

- **p. 7, l. 11: The values given are at odds with what is written p. 5, l. 12, concerning G, $\lambda$, and ECS. Please, give an explanation (which is probably to be found in the fact that no actual equilibrium has actually been reached).**

[Figure]

Reply: New information was added in Methods section (p. 5 l. 26-30)

- **p. 7, l. 13: "... similar to those of Andrews et al. ..."; in fact, the reader certainly expects no less than this, as those authors used CMIP5 data as well. Where does the difference come from? Interpolation as mentioned on p. 4, l. 23?**
Reply: Small differences are found in the analysis, which we attribute to the interpolation of the data. It was better explained in the manuscript (p.8 l. 3-8)

- **In the simulations with BESM, has there any form of "radiation double calling" been used to calculate radiative forcings or feedbacks? That could help to assess whether the radiation parameterization within BESM produces results (largely) consistent with the GFDL and NCAR kernels.**
Reply: There is not a "radiation double calling" module in BESM. We agree that it is could be a important implementation to include in future versions.

- **p. 7, l. 28: "Both radiative kernels are used ..."**
Reply: Done.

- **p. 8, l. 4: The following discussion (of Figure 4) is an example in a text flow that is largely out of scope with the paper's focus. Most of this is established knowledge from a multitude of previous papers. A clear change of perspective towards the specific features of BESM is advisable.**
Reply: The figure description was change to include the new perspective requested (p. 8 l. 29)

- **p. 8, l. 6: "The faster increase ...", I assume you mean "stronger", don't you?**
Reply: It was changed to "stronger" (p. 9 l. 35).

- **p. 8, l. 7: The two sentences discussing the possible cause-and-effect**

[Figure]

relation of water vapor and lapse-rate feedbacks is somewhat confus-
ing. The general notion, I think, is the different degree of turbulent
mixing in tropical, mid and polar latitudes. I recommend referring to,
e.g., Po-Chedley et al. (2018), who draw a lucid and consistent picture
of the latitudinal differences.
Reply: A paragraph was changed to include a more clear explanation (p. 9
l. 1-11).

- **p. 8, l. 16: "... as noted in yellow and blue shaded areas in Figure 4",
this hint would better be given when the discussion of Figure 4 starts
(l. 4) or, alternatively" in the figure caption.**
Reply: It was removed.

- **p. 8, l. 22: This paragraph is either too short (different cloud feedback
results from different methods being a highly complex issue) or too
long (as these general issues are not necessarily within the scope of
the paper). Please focus on what could be a reason for the specific
behavior of BESM in this particular case.**
Reply: Parts with comparison between regression and kernel methods was
removed.

- **p. 8, l. 35: "This is due to ...", a rather technical reasoning (which
continues throughout this paragraph). The reader would rather be in-
terested in the physical reason. I the cloud cover response over sea
(60°S) and over sea ice (Arctic) less well simulated by BESM compared
to land areas? Or could it by that there is a problem with the cloud
phase feedback (e.g., Mitchell et al., 1989; Tan et al., 2016) in BESM? I
would find it sufficient, if some ideas could be formulated, with hints
to future research.**
Reply: (p. 9, l. 29) It is evident from figures presented in the manuscript, that
BESM is an outlier for the cloud feedbacks. This is due to a strong short-
wave component response over both the Arctic and the Southern Ocean

near Antarctica. Considering the SW CRE/$\Delta T_{as}$ [as described by Cess et al. (1989)] and the individual components of feedbacks cloud mask, we can note that those higher values cloud feedback are mainly consequences of the sum of SW CRE/$\Delta T_{as}$ and the cloud masking for albedo feedback [-$(\lambda_a - \lambda_a^c)$], as shown in Figure 1. For Arctic region, the major contributor for BESM be an outlier is the SW CRE, while for over the ocean near the Antarctic is the albedo feedback cloud mask. In this latter, since the radiative kernel for both all- and clear-sky are the same throughout the models, the difference among them is due to the albedo change [$\Delta a/\Delta T(K_a - K_a^{cs})$]. Over the both regions (Arctic and near Antarctic), an increase in cloud fraction above 850 hPa and a decrease below such level for BESM is observed, which means a low-level clouds upward shifting. Moreover, the increase in cloud cover above 850 hP is stronger than the reduction below principally over the (Figure 2a). As consequence, a negative SW CRE change is present in those regions (but not stronger for BESM comparatively to other models), that is the response to the increase in sun shading (Figure 2b). However, the SW cooling is smaller than the heating provided by LW radiation, as presented in the net effect (Figure 2d). The net radiation heating change is more intense around 60°S, that can be related to the more intense surface albedo change as well as the low-cloud lifting.

- **p. 8, l. 6: "stratocumulus region", this is presumably a different entity and not connected to the BESM peculiarities showing up in Figure 4.** Reply: This part was deleted.

- **p. 9, l. 10: The section 4.3 with its figures 6 and 7 (the scatter plots) is not very insightful to me. What are these correlation diagrams (especially Figure 6) supposed to teach the reader? Is this a standard diagnostic? Does the placement of BESM in the third quadrant reveal anything about this model in a physical sense? Please, give some reasoning for the figure's usefulness in the present paper. Interpretation**

[Figure]

of precipitation change patterns is more lucid; yet, it would be fine to
know whether, e.g., the southward shift of the SPCZ in BESM does oc-
cur in other CMIP models too (even if not in the ensemble mean).
Reply: The diagrams helps understand the models temperature and precip-
itation dispersion. Moreover, it was used to answer if there is some general
behaviour, such as: Do warmer/wetter models in piControl run present also
a warmer/wetter in the abrupt4xCO2? Are there some physical limitations?
Maybe the way it was presented is not clear, so we decided rewrite the para-
graph.

- **p. 9, l. 27: "... near the equator compared to the subtropics ..."; ("as
  opposed to" suggests that the subtropics grow colder)**
  Reply: Done.

- **The statement beginning on p. 10, l. 21 "This increase ..." sounds
  somewhat counter-intuitive and is, in my opinion, an oversimplifica-
  tion of what the cited papers actually say. Rather, the non-linear in-
  crease of water vapor available for condensation, as suggested by the
  Clausius-Clapeyron relation, is limited towards a more linear relation
  by tropospheric radiative cooling (Mitchell et al., 1987).**
  Reply: New information was provided in order to make the sentence clear (p.
  11 l. 26): "The slope of the linear regression is 2.5% of precipitation change
  per K. This is a value close to that found by Held and Soden (2006).This
  slope is much inferior to that expected for Clausius-Clapeyron relation, which
  is about 6,5% of precipitation chang per K. In fact, precipitation increasing
  is not governed by the availability of moisture but by the surface and tropo-
  spheric energy balance (Allen and Ingram, 2002, Mitchell et al. 1987)."

- **p. 10, l. 25: "ACCESS1-0 and HadGEM2-ES use ..." up to the end of
  this paragraph: that may all be true, but the reader would rather be
  interested whether this implies anything for BESM.**

[Figure]

Reply: Some discussion about ACCESS1-0 and HadGEM2-ES were suppressed.

- **p. 10, l. 32: "... (SLP) pattern ..."**
  Reply: Done.

- **p. 10, l. 30: This whole paragraph gives a lot of (by no means unfounded!) physical reasoning on tropospheric variability patterns, but in the end takes a simple similarity of the SLP mean response patterns from BESM and from the CMIP ensemble to indicate that BESM may well represent such variability patterns. This is a bold conclusion, which in my view would need backing from actual variability pattern analysis. Is such analysis planned?**
  Reply: The comparison between BESM and ensemble was removed. The BESM variability change is planned to be discussed in a future work.

- **p. 11, l. 23: "It is shown ...", this a very odd 'conclusion', as this statement is common knowledge motivating any research on global warming, and it is certainly not "... shown in this study". Even "... confirmed by this study" would be a summary much too weak for motivating publication of this paper. Please, find a more specific main conclusion that is directed towards the BESM performance.**
  Reply: New information was added.

- **p. 11, l. 31: Here, some information about the BESM radiation module and its evaluation would emphasize that the radiative feedbacks calculated from BESM output within the CMIP range indeed indicate a good representation of such feedbacks inside that model (see major issues).**
  Reply: New information was added.

- **p. 12, l. 5: You might delete "However,"; I see no contradiction of this sentence with the preceding one.**
  Reply: Done.

[Figure]

- **p. 12, l. 12: "... is not the aim for the BESM development", this whole paragraph is a very puzzling wrap-up of your paper (see general remarks).**
  Reply: New information was added.

- **Figure 3, Figure 8: Please ensure that this figure will appear larger in the eventual paper, otherwise it will be hard to decipher.**
  Reply: Done.

- **Caption of Figure 6: "Shaded areas"; this return in several other figure captions, too. You mean the white areas, don't you?**
  Reply: We mean areas fill in with colors.

**Complete Figure Captions**

[revised manuscript text omitted]

15   W m$^{-2}$K$^{-1}$. This result is similar to that found by Soden et al. (2008) for CMIP3 [IPCC AR4, IPCC (2007)] models, where they presented a near neutral and positive cloud feedback. ΔCRE computed by using the Gregory et al. (2004) methodology (Section 4.1) is related to the cloud feedback (estimated from corrected ΔCRE - Section 4.2), even though ΔCRE and $\lambda_c$ could present opposite signals for some models. For instance, BESM shows -0.13 BESM presents positive values of around 0.5 W m$^{-2}$K$^{-1}$ and 0.95 for both SW and LW cloud feedback, which results in a total cloud feedback of 1.0 W m$^{-2}$K$^{-1}$ for

20   ΔCRE and $\lambda_c$, respectively. (Figure 3). The highest positive values are in regions with strong albedo feedback (Figure 4).

    Overall, BESM lies within the range of CMIP5 models, with global-mean values of 1.24 W m$^{-2}$K$^{-1}$, 0.95 W m$^{-2}$K$^{-1}$, 0.27 W m$^{-2}$K$^{-1}$, -3.57 W m$^{-2}$K$^{-1}$ and -0.71 W m$^{-2}$K$^{-1}$ for water vapor, cloud, albedo feedbacks, Planck and lapse-rate feedbacks, respectively. However, differences between BESM and the other models are found in the high latitudes, where BESM exhibit lapse-rate and humidity feedbacks marginally out of range of values set by the CMIP5 multi-model ensemble

25   (Figure 4). It is also evident from figures 4 and 5 that BESM is an outlier for the cloud feedbacks. This is due to a strong shortwave component response over both the Arctic and the Southern Ocean near Antarctica. Even though BESM presents an acceptable radiation closure at TOA (less than 2 W m$^{-2}$ bias), it is deficient in representing shortwave radiation over the Arctic and the Southern Ocean, as a result of negative biases of middle and high clouds in the extratropics as shown by Casagrande et al. (2016). Therefore, it is plausible that such a deficiency in cloud representation could cause the high positive

30   values in SW cloud feedback . It is worth highlighting that the cloud feedback is one of the biggest causes of uncertainty in climate projections (IPCC, 2013). BESM has also a larger negative cloud feedback in the stratocumulus regions compared to the CMIP5 ensemble (figures 4 and 5). This, combined with its anomalous positive high latitude cloud feedback, is qualitatively consistent with the results presented in McCoy et al. (2016) showing an anti-correlation across models in Con
[revised manuscript text omitted]

McCoy, D. T., Tan, I., Hartmann, D. L., Zelinka, M. D., and Storelvmo, T.: On the relationships among cloud cover, mixed-phase partitioning, and planetary albedo in GCMs: CLOUD COVER, MIXED-PHASE, AND ALBEDO, Journal of Advances in Modeling Earth Systems, 8, 650–668, https://doi.org/10.1002/2015MS000589, 2016.

Mellor, G. L. and Yamada, T.: Development of a turbulence closure model for geophysical fluid problems, Reviews of Geophysics, 20, 851, https://doi.org/10.1029/RG020i004p00851, 1982.

Miller, R. L., Schmidt, G. A., and Shindell, D. T.: Forced annular variations in the 20th century Intergovernmental Panel on Climate Change Fourth Assessment Report models, Journal of Geophysical Research, 111, https://doi.org/10.1029/2005JD006323, 2006.

Morrison, H., Curry, J. A., and Khvorostyanov, V. I.: A New Double-Moment Microphysics Parameterization for Application in Cloud and Climate Models. Part I: Description, Journal of the Atmospheric Sciences, 62, 1665–1677, https://doi.org/10.1175/JAS3446.1, 2005.

Nobre, P. and Srukla, J.: Variations of Sea Surface Temperature, Wind Stress, and Rainfall over the Tropical Atlantic and South America, Journal of Climate, 9, 2464–2479, https://doi.org/10.1175/1520-0442(1996)009<2464:VOSSTW>2.0.CO;2, 1996.

Nobre, P., Siqueira, L. S. P., de Almeida, R. A. F., Malagutti, M., Giarolla, E., Castelão, G. P., Bottino, M. J., Kubota, P., Figueroa, S. N., Costa, M. C., Baptista, M., Irber, L., and Marcondes, G. G.: Climate Simulation and Change in the Brazilian Climate Model, Journal of Climate, 26, 6716–6732, https://doi.org/10.1175/JCLI-D-12-00580.1, 2013.

Pincus, R., Forster, P. M., and Stevens, B.: The Radiative Forcing Model Intercomparison Project (RFMIP): experimental protocol for CMIP6, Geoscientific Model Development, 9, 3447–3460, https://doi.org/10.5194/gmd-9-3447-2016, 2016.

Pithan, F. and Mauritsen, T.: Arctic amplification dominated by temperature feedbacks in contemporary climate models, Nature Geoscience, 7, 181–184, https://doi.org/10.1038/ngeo2071, 2014.

Plass, G. N.: The Carbon Dioxide Theory of Climatic Change, Tellus, 8, 140–154, https://doi.org/10.1111/j.2153-3490.1956.tb01206.x, 1956.

Richter, I., Xie, S.-P., Behera, S. K., Doi, T., and Masumoto, Y.: Equatorial Atlantic variability and its relation to mean state biases in CMIP5, Climate Dynamics, 42, 171–188, https://doi.org/10.1007/s00382-012-1624-5, 2014.

Shell, K. M., Kiehl, J. T., and Shields, C. a.: Using the radiative kernel technique to calculate climate feedbacks in NCAR's Community Atmospheric Model, Journal of Climate, 21, 2269–2282, https://doi.org/10.1175/2007JCLI2044.1, 2008.

5  Soden, B. and Held, I.: An Assessment of Climate Feedbacks in Coupled Ocean – Atmosphere Models, Journal of Climate, 19, 3354–3360, https://doi.org/10.1175/JCLI9028.1, 2006.

Soden, B. J., Broccoli, A. J., and Hemler, R. S.: On the use of cloud forcing to estimate cloud feedback, Journal of Climate, 17, 3661–3665, https://doi.org/10.1175/1520-0442(2004)017<3661:OTUOCF>2.0.CO;2, 2004.

Soden, B. J., Held, I. M., Colman, R., Shell, K. M., Kiehl, J. T., and Shields, C. A.: Quantifying Climate Feedbacks Using Radiative Kernels,
10  Journal of Climate, 21, 3504–3520, https://doi.org/10.1175/2007JCLI2110.1, 2008.

Tarasova, T. a. and Fomin, B. a.: The Use of New Parameterizations for Gaseous Absorption in the CLIRAD-SW Solar Radiation Code for Models, Journal of Atmospheric and Oceanic Technology, 24, 1157–1162, https://doi.org/10.1175/JTECH2023.1, 2007.

Taylor, K. E., Stouffer, R. J., and Meehl, G. a.: An overview of CMIP5 and the experiment design, Bulletin of the American Meteorological Society, 93, 485–498, https://doi.org/10.1175/BAMS-D-11-00094.1, 2012.

15  Thorpe, L. and Andrews, T.: The physical drivers of historical and 21st century global precipitation changes, Environmental Research Letters, 9, 064 024, https://doi.org/10.1088/1748-9326/9/6/064024, 2014.

[revised manuscript text omitted]

---

## Referee Report (RR1)

Review of "Assessing the performance of climate change simulation results from BESM-OA2.5 in comparison to a CMIP5 model ensemble" by V.B. Capristrano et al. (revised manuscript)

**Overall assessment and recommendation**

I regret to conclude that this paper has not been sufficiently improved by the revision process to be acceptable. While I appreciate that the authors have tried to bring the characteristics of BESM-OA2.5 more in focus of their presentation (rather than discussing the general performance of CMIP5 models), the results is still a clumsy and partly dis-organized concatenation of results and result comparisons that do not lead to a clear assessment of the suitability of BESM for specific purposes. New text often has been insufficiently harmonized with the previous text, making reading through the manuscript still an extremely arduous task.

As I stated in my original review, my impression is that BESM is a reasonable model that could be useful for specific applications at least. Hence, I am reluctant to reject this paper once and for all. The authors should be allowed to make one more attempt to create a straightforward paper with a coherent message. To this end (as I have proposed before) the focus of future use of BESM should be made clear, considering the merits and shortcomings of this model. The authors should intensify their attempts to interlink the evidence arising from individual parameter evaluation. This already has been tried in a number of cases, but it too often results in circular reasoning, not approaching the roots of characteristic BESM features. Finally, I emphasize that just executing through my list of technical and language suggestions alone will not do! The author team apparently does not include an English native speaker, hence assistance in producing a proper English text ought to be given by either the editorial office or from some other consultant. Otherwise, I fear that I will be reluctant to read through this paper once again.

**General remarks**

- 1) Section 2.1 still contains elements of a comparison between BESM-OA2.5 and BESM-OA2.3 (e.g., p. 3, 1.32) though a dedicated section (2.2) is supposed to cover such differences.
- 2) It is on occasions still hard to reproduce what has actually been done and why (e.g., p. 6, l.25).
- 3) No reason is given on p. 8, 2nd paragraph, why only 11 rather than 15 CMIP5 models are included here. Or are sometimes 11, sometime 15, models used, as could by read out of p. 8, l. 6?
- 4) Occasionally, I still miss a comment on the specific performance of BESM, even if it's well consistent with the CMIP5 ensemble (e.g. Figure 3).
- 5) Page 9, 1st paragraph: This has been reformulated, but is now even more confusing than before. Please reconsider, what is the intended message

here, with focus on BESM. Then stick to specific reasoning to underpin that message.

- 6) Page 9, 2nd paragraph: Here, too, the line of reasoning remains badly organized: What is the message: Does BESM simulate a stronger Arctic amplification than the CMIP ensemble (suggested by the more negative Planck feedback)? This could simply explain more snow/ice melting. Evidently the lapse rate feedback in BESM is exceptionally positive at Arctic latitudes, pointing at a enhanced vertical gradient in the temperature response. Can this be discussed in the context of the Veiga et al. paper (atmospheric temperature response)?
- 7) The last paragraph of section 4.2, with much newly introduced text, is very hard to understand both concerning the weak use of English language and a confusing inherent logic. I have read through this paragraph three times, but then gave up, being unable to reconcile the statements in the text with what the figures display.
- 8) Scatter Diagrams in Figures 8 and 9: Do you conclude anything from the apparent correlation between precipitation in piControl and abrupt4xCO2 on one side, and missing correlation for respective surface temperature levels on the other side? Does this have implications for the BESM model performance.
- 9) In the last paragraph of the conclusions an outlook to what is planned with BESM-OA2.5 (future research focus) is still lacking. However, this would be the logical outcome of the assessment of its merits and shortcomings, which I assume is what the present paper has been written for.

**Language and Technical Remarks**

p. 1, l. 7 (Abstract): " ... the CMIP5 ensemble mean value ... "

p. 1, l. 8 (Abstract): " ... BESM simulation show zonally average feedbacks, estimated from radiative kernels, that lie within the ensemble standard deviation ..."

p. 1, l. 11 (Abstract): "... BESM also features a strong positive ..."

p. 1, l. 12 (Abstract): As this sentence mentions a merit of BESM, while the preceding sentence comments on a disagreement with CMIP, "moreover" makes quite an unlucky connection. By the way, "consistent" with what?

p. 2, l. 7: "... results in a temperature rise ..."

p. 3, l. 7: "... models, also discussing peculiarities in the BESM climate response."

p. 3, l. 16: "... same as used by Veiga ..."

p. 3, l. 17: "... model, with its dynamical core being based on ..."

p. 3, l. 22: "... of physical parameterizations between BAM (as used in this paper) and BAM NWP ..."

p. 3, l. 24: "... 28 layers, unevenly spaced, in the ..."

p. 3, l. 29: "... is able to capture ..."

p. 3, l. 30: "... with a double ITCZ ..."

p. 3, l. 31: "improvement", despite of the "substantial biases" addressed in the preceding sentence?

p. 3, l. 33: "... decadal climate variability patterns." This is meant, isn't it?

p. 4, l. 4: I understand that AMOC is a circulation structure rather than a parameter. So, what "value" a you referring to? If required, please give an absolute or relative difference of the parameter you have in mind.

p. 4, l. 9: "... are determined, which are important ...". Anyway, the content of this sentence to me resembles what is given below (p. 4, l. 14), with the sentences in between (starting with "The total energy balance ...") causing an awkward logical break.

p. 4, I 19: This sentence again repeats what is given in p. 4, I. 9 ...

p. 5, l. 6: "... which means a spin-up of 150 years." Does this mean that the 150 yrs of abrupt4xCO2 are regarded as a spin-up here (due to their non-equilibrium character)? Or are 150 yrs of abrupt4xCO2 swapped as a spin-up, and *another* 150 yrs evaluated as some kind of quasi-equilibrium? Please, clarify.

p. 5, l. 6: "... commonly employed ... for climate change assessment"; please, be careful to distinguish between "climate change assessment" and "climate sensitivity assessment"! In my view, "climate change" in the CMIP context is rather assessed through historical simulations and future scenario simulations.

p. 5, l. 12: In this paragraph the "forth and back" jumping in addressing the merits of the regression and kernel method is somewhat confusing but could be easily avoided.

p. 5, I 27, 28: There's still something wrong with the sentences here. Suggestion: "As G can be approximated by backward regression towards  $\Delta T_{as}=0$ , ECS can be estimated as ECS= $-G/\lambda$ ."

p. 5, I 30: "... it is common to divide the result derived from 4xCO2 simulations by 2 (Andrews ..."

p. 6, l. 9: "... is used next, in order to partition the ...", as "next to" is confusing. By the way, "separate" or "split" may be preferred to "partition".

p. 6, l. 14: "integraly" -> "fully" (or "necessarily")

p. 6, l. 16: "This, however, assumes that ..."

p. 6, equation 3:  $T_{as}$  is the near surface temperature (p. 5), but what is then  $T_s$ ? I tried to clarify this by looking into Vial et al. (2013), without success. Please, be precise in citing, or explaining what you have done, and why.

p. 6, l. 25: Confusing: As q is in the data base, why should it be approximated based on the assumption of constant relative humidity? To my knowledge, this is not common in feedback analysis. Is it possible that you are mis-interpreting the cited references here?

p. 7, l. 5: "... changes are not accounted ..."

p. 7, equation 5: It is not immediately obvious, what the indices "a" and "k" mean.

p. 8, l. 4: "... were assessed as it was performed ..."; I do not understand this sentence. Are the data not from the ESGF data base (p. 5, l. 10) ??

p. 8, l. 6: "... e inmcm4 ", did you intend "... and inmcm4"?

p. 8, l. 13: witch -> which

p. 8, l. 29: Please, explain how Figure 4 is related to Figure 3. Is it simply an average over the latitudinal profile of Fig. 3? Your discussion of the Planck feedback is casting doubts concerning this: If it's constant by about -4 Wm-2K-1 (l.29) with mostly negative deviations at polar latitudes, how can this result in a global mean of -3.6 Wm-2K-1 (l. 26)? Please, cross-check the numbers.

p. 8, l. 32: "stronger vertically homogeneous warming". This is a strange reasoning, as the Planck feedback is essentially the surface warming, constantly extrapolated upward through the depth of the troposphere. Can the message of this sentence be reconciled with Figure 8?

p. 9, l. 20: The partly revised text in this paragraph (see also major comments) contains some sensible elements, but is also moving in circles, explaining stronger sea-ice melting with stronger surface warming and vice versa. More reorganisation of the text is necessary.

p. 9, l. 22: "Those negative values ...", it is unclear which values are addressed.

p. 9, l. 30: "The highest positive values ...", I would expect that backscattering increases if ice turns into water, driving the shortwave cloud feedback to *more negative* values. However, your later discussion (Figure 7, see also below) seems to suggest that the longwave cloud feedback is the dominant component.

p. 9, l. 31: I feel that the following text (until "... outlier for the cloud feed-backs.") is mainly repetitive.

p. 10, l. 4: " $\lambda_a$ ,  $\lambda_{ac}$ ", are you referring to an analytical framework that is given in Cess et al. (1989)? Otherwise the reader is rather left in the dark here.

p. 11, l. 4: "models with ... apparently do not show ..."; please also replace "present" by "show" on many occasions thereafter.

p. 11, l. 24: "...quadrupling of atmospheric  $CO_2$  with the piControl pre-industrial  $CO_2$  concentrations ...": meaning what? The two first sentences of this paragraph appear to transport the same statement.

p. 11, l. 28: "... precipitation increase is not governed ..."

p. 11, l. 31: Does this have in any way implications for the use of these somewhat "outlying" models?

p. 12, l. 13: "...regions with the strongest increase of westerly winds at all levels indicate a southward jet displacement ..."

p. 12, l.18: Is "omega" something different from "vertical velocity"? Anyway, "omega" isn't self-explaining, so please adjust the text.

p. 12, l. 31: "... radiative code transference ...", do you mean "performance"? Is there any indication of that particular feature for BESM's radiative transfer model?

p. 12, l. 31: "... rapid adjustments ..."; the rapid adjustment process is included in the CMIP5 model results as well, per construction. You apparently did not calculate the rapid adjustments for BESM, but do you have any indications that there might be a systematic bias with respect to CMIP (see Smith et al., 2018).

p. 13, l. 4: "Two regions indicate enhanced inter-model standard deviation for Planck, lapse-rate and albedo feedback"; also in the rest of this paragraph the use of English language is very weak, making the meaning nearly incomprehensible for me.

**References (only if not already cited in the paper):**

Smith, C.J. et al., 2018: Understanding rapid adjustments to diverse forcing agents, Geoph. Res. Lett. 45, 12023-12031.

---

## Referee Report (RR2)

Review of "Assessing the performance of climate change simulation results from BESM-OA2.5 in comparison to a CMIP5 model ensemble" by V.B. Capristrano et al. (revised manuscript)

Overall assessment and recommendation

This manuscript has been considerably improved in response to the first review process. In particular, use of English language has been thoroughly upgraded by external consultants and - as a consequence - reading to the text has become much smoother now. Given that the objective of GMD papers is proper description and evaluation of model tools rather than coming up with innovative scientific results, I find that the paper is now on a good way towards acceptance for this journal. As I already stated in the 1st review stage, I regard BESM to be a reasonable climate model that ought to get its chance for being used for specific scientific research.

Such credible progress notwithstanding, some more work on the manuscript is still necessary. First, there are still some (though not many) inconsistencies in the results presentation that require revision. In some cases this might be caused by excessively technical language that does not always help to address scientific interrelations precisely. Second, the language upgrade (though successful in an overall sense) has failed on occasions where the language editor apparently did not comprehend the scientific meaning of some text. The last three paragraphs of the conclusion section form a prototype of what I mean: The penultimate paragraph (p. 13, l. 25f) reads perfectly convincing and clear, whereas the two paragraphs surrounding it (p. 13, l. 15f and p. 14, l. 1f) are still very hard to grasp. Language polishing thus forms the bulk of my remaining recommendations and correction requests. This time, however, it will be sufficient if the authors carefully consider the list of proposed changes.

General recommendations

1) p. 1 l. 3: I recommend to extend to "… precipitation, atmospheric circulation, and radiative feedbacks", given the room devoted to feedback analysis.

2) p. 2 l. 28: "… uncertainty in estimates of total precipitation due to uncertainties in …"; Could it be preferable to replace "uncertainty" by "inter-model deviation"?

3) p. 6 Eq. 4: Despite the statements made in your reply I am still at a loss to detect your equation (4) in any of the given references. Particularly the splitting with respect to $T_s$ and $T_{as}$.

4) p. 6 l. 24: " … they are shifted to the residuum." Concerning stratospheric temperature and the corresponding rapid radiative adjustment, this must make up for a quite large residuum then (see Smith et al., 2018, their Fig. 3). However, as you are not using the kernel derived feedbacks for closing the forcing vs. feedback balance, I will not insist on this point any further.

5) p. 7 l. 2: "This approach, however, assumes …" – I do not understand what is implied by this sentence. Does the following proposed text meet your intention? "However, the resulting kernel derived feedbacks can only be assumed to reflect the actual feedback in the considered models under the premise of small differences between the radiative transfer codes. This has not been verified here, thus enabling inconsistencies between feedbacks derived from the regression method and the kernel method. On the other hand, Figure 3 indeed suggests small differences at least arising from use of either the NCAR or the GFDL radiative transfer model."

6) p. 8 l. 12: "Thus the ratio of …"; as it sounds somewhat trivial that $1/\lambda$ (the "climate sensitivity parameter") explains ECS (the "equilibrium climate sensitivity") well, you might consider a change to: "Thus inter-model variations in the balance of feedbacks explains the dispersion in the ECS better  than …"

7) p. 8 l. 22: " … the inter-model differences are greater than the distribution of the radiatively active constituents of the base model"; This sentence is rather confusing: Which differences? Which constituents? I suspect that you intend to address the issue of point 5 above, but it's not very clear.

8) P. 9 l. 2: "… that BESM does not have a higher contrast between the surface and upper troposphere temperatures in comparison to temperature contrasts of the other models."; a still quite clumsy sentence and also counter-intuitive, as in case BESM shows a more positive lapse rate feedback than the ensemble, this should imply that the warming is relatively larger at the surface and relatively weaker at the upper troposphere, resulting in an in fact more strongly increased (!) vertical temperature gradient. That seems to contradict your statement. Do you agree? Please, reformulate accordingly.

9) **Major point!** The interpretations starting at p. 10, l. 4, with respect to Fig. 6, still lack clarity and also inherent consistency. First, it is not obvious how the term 'cloud mask' is to interpret physically, as it is not mentioned when the 'adjusted cloud radiative effect', $\Delta CRE_a$, is introduced by Eq. (5). As a consequence, the link between the analytic explanation in the penultimate paragraph of 4.2 and the discussion of physical parameters in the last paragraph remains obscure. In particular (my second main point) it is not clear to me how (l. 15) "Because of the increase in total cloud fraction, a *negative* SW CRE appears …" can be reconciled with the SW cloud feedback in Fig. 5 (mid right panel) being obviously *positive* in those very regions. Does the lower troposphere cloud response in BESM compete with the sea ice retreat differently than in the CMIP ensemble? While a full explanation is not required in the framework of a GMD paper, still you should try to explain the parameter inter-relations in a consistent and lucid way. Especially, because the concluding section resumes this point to emphasize it as a main part of the paper.

10) p. 13 l. 20: "… increase in cloud cover … which increased the outgoing SW radiation at the TOA." ; see previous point 9! In Fig. 5 the SW cloud feedback in that latitude range is positive, which suggests that more SW energy remains in the climate system, thus indicating *less reflection*. By the way, "outgoing SW radiation" ought to be replaced by "reflected solar radiation".

11) p. 13 l. 3: " … overcome by the albedo feedback cloud mask …" ; also related to major point 9. This explanation is not suggestive and not adequately pointed out in a physical sense.

12) p. 14 l. 2: "In this sense it has contributed …" You obviously refer to that part of the manuscript I have addressed in my general points 9-11. Evidently, I do not agree to this claim as not convincing argument for a "better understanding" has been provided. Nor do I demand or expect, that such a contribution to better understanding is given in the framework of a GMD paper. Hence, the statement could simply be discarded, or modified to something like " … the analysis methods used here have the potential to explain remaining process uncertainties causing inter-model spread in the cloud feedback in future work."

Language and Technical Remarks

1) p. 1, l. 7 (Abstract): " … the BESM simulations yield …"

2) p. 3 l. 10: "… physical processes …" → "… physical parameterisations …" ; "… those used by Veiga …" → "… those discussed in Veiga …"

3) p. 3 l. 12: " … is used in this study but with a …" → "… is generally used in BESM-OA2.5, except for a …"

4) p. 3 l. 29: "… the value observed by the RAPID project …" → "… the value determined within the RAPID project …"

5) p. 4 l. 1 (heading): "Comparison to a previous version" → "Comparison to a previous model version"

6) p. 4 l. 2: "evolution" → "advancement"

7) p. 4 l. 5: " …, which results in … to approximately –4 Wm$^{-2}$ …" → "…, with a reduced global mean bias of approximately –4 Wm$^{-2}$ …"

8) p. 4 l. 26: "… presented less consistent results" → "yielded less consistent results"

9) p. 5 l. 1: "… both experiments were run in parallel for 150 years." → "… for years 151 to 300 both scenarios are run in parallel to the piControl reference for 150 years." [I understand that for the evaluation of piControl, e.g. means of the reference state, only the last 150 years of piControl have been used. If, however, reference means are averaged over 300 years, this should be mentioned explicitly in the text.]

10) p. 5 l. 7: "… two different methods: regression (Gregory et al., 2004) and radiative kernel …" → "two different methods, using either a regression according to Gregory et al. (2004) or radiative kernels …"

11) p. 5 l. 14: "… method consists of the …" → "… method consists in the …" [or "… method involves the …"]

12) p. 5 l. 15: " … the net radiation change …" → "the net radiative flux change …"

13) p. 6 l. 1 (heading): "Climate feedbacks (radiative kernel)" → "Separating individual climate feedbacks using radiative kernels"

14) p. 6 l. 7: " … kernels consist of the impact in the radiative balance in the TOA via arbitrary increases in the …" "… kernels represent the impact on the radiative balance at TOA via arbitrary increases of the …"

15) p. 6 l. 8: "For *calculating* the temperature kernel, *an increment of* 1 K is added …"

16) p. 6 l. 9: "*For* the albedo kernel …"

17) p. 7 l. 1: "the models analyzed" → "the CMIP model analyzed"

18) p. 7 l. 23: "These linear regressions …"; I recommend the following formulation of this sentence as an improvement: "The linear regressions based on all-sky radiative fluxed are used to estimate ECS, *G* and *λ*, while the regressions based on clear-sky data are used to obtain ΔCRE …"

19) p. 8 l. 2: "… were assessed as previously performed by Andrews et al. (2012) for 15 …" → "were compiled in extension of the previous work by Andrews et al. (2012), who evaluated 15 …"

20) p. 8, l. 4: "For the 15 models …"; I recommend to reformulate to: "In Andrews et al. (2012) the ECS ranges from 2.07 to 4.74 K for the 15 models analyzed there, which is largely confirmed by our analysis."

21) p. 8 l. 5: "The possible small differences can be attributed …" → "The small differences can possibly be attributed …"

22) p. 8 l. 14: "… that is *well* within the range of …" [certainly an argument in favor of your BESM].

23) p. 8 l. 25: "The Planck feedback global-mean is negative …" → "The global mean Planck feedback is [strongly] (?) negative …"

24) p. 8 l 28: "… which is an overestimation compared to the ensemble mean value" → " … slightly underestimating the ensemble mean value in magnitude"; Technically, your formulation is of course correct, but "overestimation" is somewhat counter-intuitive to suggest that it is in fact "less negative".

25) p. 8 l. 31: I recommend adding the following introductory sentence: "Figure 4 shows the latitudinal profiles basic to the global mean feedback values of Figure 4, allowing to identify the regions that induce deviations of BESM results from the CMIP ensemble. In Figure 4a-b, there is …"

26) p. 8 l. 33: "… with nearly the same increased …" → "… with a similarly increased …"

27) p. 8 l. 34: "… one of the lowest values …" → "… one of the most negative values …" [see point 24 above]

28) p. 9 l. 1: "(with respect to the surface temperature)" → "(corresponding to large surface warming)"

29) p. 9 l. 5: "adiabate"

30) p. 9 l. 6: "… is greater at the upper troposphere …" → "… is larger in the upper troposphere …"

31) p. 9 l. 8: " … is responsible for a potentialization of the greenhouse gas effect, revealing a positive …" → "… is causing a reinforcement of the greenhouse effect, reflected by a positive …"

32) p. 9 l. 9: " … of this *close* link …"

33) p. 9 l. 10: " … it is common to sum their effect, as performed in …" → "… it is common to consider their effects as the sum, as displayed in …"

34) p. 9 l 11: "The greatest BESM deviations … for all models"; Does the message of this statement differ from what has been told on p. 9, l. 2-4 ? [ see also general remark No. 8) If not, my recommendation is to discard this sentence.

35) p. 9 l.16: "The albedo feedback values computed for BESM and other CMIP models are show in …" "The albedo feedback profiles from BESM and the CMIP models are compared in …"

36) p. 9 l. 16: "These results are particularly important over …"; somewhat fuzzy, better change to "Non-zero results mostly occur over …"; You might also consider to support this statement by citing evidence from papers that have presented corresponding geographical distributions like Chung and Soden (2015, their Fig. 9) or Block et al. (2020, their Fig. 2a). Accordingly, respective information for the Planck feedback (p. 8 l. 32) is provided by Rieger et al. (2017, their Fig. 2).

37) p. 9 l. 18: " … all of the models …" → "all models"

38) p. 9 l. 20: " … how fast the sea-ice melts in …"

39) p. 9 l. 21: "showed" → "show"

40) p. 9 l. 23: "Pithan and Mauritsen"; here, too, the recent paper from Block et al. (2020) [see point 36] could be a very appropriate reference.

41) p. 9 l. 31: "… as they reported a nearly neutral and positive …" → "…, who also reported a near-zero to positive …"

42) p. 9 l. 35: "… in the high latitudes." → "… at high latitudes."

43) p.10 l. 2: " … assess the *analytical* causes …" [to distinguish from the more physical cause addressed in the subsequent paragraph]

44) p.10 l. 5: "… all-minus clear-sky radiation …" "…all-sky minus clear-sky radiative  flux …"; However, the meaning of the sentence is hard to comprehend anyway; see general comment No. 9.

45) p. 10 l. 6: "We observe that …" → "We find that"

46) p. 10 l. 9: "… it is possible to see that the major contributor to BESM's status … is the SW CRE" → "… it is possible to attribute BESM's status … to the SW CRE …"

47) p. 10 l. 12: "A further analysis …" → "A deepened physical analysis …"

48) p. 10 l. 27: " … in the air temperature … " ??? - Do you mean "… in the surface temperature …", which is what is shown in Figure 8 ?

49) p. 10 l. 29: "shows" → "yields" (2x)

50) p. 10 l. 32: " … have similar …" → "… have closely similar …"; "… there are few spatial correspondences between …" → "… there is hardly any pattern correlation between …"

51) p. 11 l. 4: "…, which show …" → "…, who show …"

52) p. 11 l. 7: " … mainly over the North Pole." → "… particularly over the Arctic" [certainly the North Pole as a specific point in space does not show maximum warming].

53) p. 11 l. 8: "… shows an increase in temperature …" "… shows a relative maximum in warming …" [temperature increase occurs almost everywhere!]

54) p. 11 l. 14: "… not possible to note a linear …" → "… not possible to claim any robust linear …"

55) p. 11 l. 17: "Plank" → "Planck"

56) p. 11 l. 20: "… are approximately similar to …" → "… are, in general, similar to …"

57) p. 11 l. 25: "… notable discrepancies." → "… notable exceptions."

58) p. 11 l. 31: "… suggests a linear …" → "… emphasizes a linear …" ; "indicating "

59) p. 11 l. 33: "As shown in …" → "As obvious from Figure 9, the BESM performance perfectly matches the ensemble mean behavior in the global mean."

60) p. 12 l. 4: "The apparent …" → "An apparent …"

61) p. 12 l. 19: "This SLP decrease …indicate a …" → "This, connected with the increase in the mid-latitudes, indicates a …"

62) p. 12 l. 32: "Figure 13 shows …" ; this sentence needs rectification. First, I notice no "zonal wind" in the figure. Second, your explanation in the reply notwithstanding, the difference between "omega" and "omega vertical motion" is not clear. You might consider to introduce the term of "generalized vertical velocity" (omega [pa/s]) to distinguish it from the physical vertical motion (w [m/s]). If, however, such a distinction is irrelevant for the interpretation of the figure, you might for simplicity just replace "omega" by "vertical velocity".

63) p. 13 l. 2: "… positive values …" → "… enhanced subsidence …"

64) p. 13 l. 7: " … climate sensitivity parameters …" → " … key sensitivity parameters …"

65) p. 13 l. 16: "… considerable standard deviations …" → "considerable inter-model variability …"

66) p. 13 l. 17: " … BESM also showed cloud feedback values larger than the zonal mean plus standard deviation for the analyzed models …" ; that's not easy to comprehend, do you mean " … the BESM zonal mean cloud feedback ranges outside the standard deviation for the analyzed models …" ?

67) p. 13 l. 19: "… was approximately 0 Wm$^{-2}$K$^{-1}$." → "… was close to zero."

68) p. 13 l. 32: " … was clearer in … "→ "… was more distinct in …"

69) p. 14 l. 1: "BESM is …" "Summarizing, we conclude that BESM-OA2.5 is a climate model that can reproduce approved physical processes that determine and modify changes of the global climate system."

70) p. 14 l. 4: " … the BESM development team is committed to improving the cloud cover of the model as well as …" → "…  the BESM team continues its effort to improve the cloud parameterization of the model as well as …"

71) p. 14 l. 5: " … is under revision." That could easily used as an argument to reject the paper. Hence, I recommend " … is seen as an issue that is to be tackled in ongoing model development work."

72) p. 14 l. 6: "… it will be more compatible … ocean an atmosphere." → "… it will include physical parameterizations of atmosphere/ocean interaction that lead to better agreement with other model and with observations."

73) Figure 5: Are these feedbacks as the figure caption suggests? Then the label of the colour bar should be "$Wm^{-2}K^{-1}$".

74) Figure 6: There is no indication of physical units in that figure or figure caption.

75) Figure 7: There is no indication of physical units in that figure or figure caption.

76) Figure 8, caption: "…isoline…" → "contour line"

77) Figure 9, caption: "…isoline…" → "contour line"

78) Table 3, caption: $\Delta CRE$ ought to have unit $Wm^{-2}$.

References (only if not already cited in the paper):

Block, K. et al., 2020: Climate models disagree on the sign of total radiative feedback in the Arctic, Tellus A, 72, 1696139.

Chung, E.-S. and Soden B.J., 2015: An assessment of direct radiative forcing, radiative adjustments, and radiative feedbacks in coupled ocean-atmosphere models, J. Clim., 28., 4152-4170.

Rieger, V.S. et al., 2017: Can feedback analysis be used to uncover the physical origin of climate sensitivity and efficacy differences? Clim. Dyn., 49, 2831-2844.

Smith, C.J. et al., 2018: Understanding rapid adjustments to diverse forcing agents, Geoph. Res. Lett. 45, 12023-12031.

---

## Author Response (AR3)

**Reply to the GMD anonymous reviewers**

First of all, I would like to thank the two anonymous reviewers. They contributed a lot with the improvement of this final version the manuscript. Below is found the point-by-point reply to the comments of the Referee #2.

**General recommendations section:**

**1) p. 1 l.3: I recommend to extend to "… precipitation, atmospheric circulation, and radiative feedbacks", gin the room devoted to feedback analysis.**
Reply: ok, we totally agree. (p. 1 l. 3)

**2) p.2 l.28: "… uncertainty in estimates of total precipitation due to uncertainties in ...", Could it be preferable to replce "uncertainty" by "inter-model deviation"?**
Reply: yes, it was changed. (p. 2 l. 28)

**3) p. 6 Eq. 4: Despite the statements made in your reply I am still at a loss to detect your equation (4) in any of the given references. Particularly the splitting with respect to Ts and Tas.**
Reply: Jonko et al. (2013) has been added to the description of Eq. 4. The Eqs. 3-5 of Jonko et al. (2013) have the splitting with respect to Ts and Tas, as requested. (p. 6 l. 13)

**4) p. 6 l. 24: "… they are shifted to the residuum." Considering stratospheric temperature and the corresponding rapid radiative adjustment, this must make up for a quite large residuum then (see Smith et al., 2018, their Fig. 3). However, as you are not using the kernel derived feedbacks for closing the forcing vs. feedback balance, I will not insist on this point any further.**
Reply: It is true. The residuum is large.

**5) p.7 l.2: "This approach, however, assumes..."- I do not understand what is implied by this sentence. Does the following proposed text meet your intention? "However, the resulting kernel derived feedbacks can only be assumed to reflect the actual feedback in the considered models under the premise of small differences between the radiative transfer codes. This has not been verified here, thus enabling inconsistencies between feedbacks derived from the regression method and the kernel method. On the other hand, Figure 3 indeed suggests small differences at least arising from use of either the the NCAR or the GFDL radiative transfer model.**
Reply: the rewrite suggestion was accepted. (p. 7 l. 2)

**6) p. 8 l. 12: "Thus the ratio of ..."; as it sounds somewhat trivial that 1/λ (the "climate sensitivity parameter") explains ECS (the "equilibrium climate sensitivity") well, you might consider a change to: "Thus inter-model variations in the balance of feedbacks explains the dispersion in the ECS better than …"**
Reply: the sentence was rewrite as indicated in the comment. (p. 8 l. 14)

**7) p.8 l. 22: "… the inter-model differences are greater than the distribution of the radiatively active constituents of the base model"; This sentence is rather confusing: Which differences?**

**Which constituents? I suspect that you intend to address the issue of point 5 above, but it's not very clear.**
Reply: following this comments we rewrite the sentence. (p. 8 l. 24)

**8) p.9 l. 2: "… that BESM does not have a higher contrast between the surface and upper troposphere temperatures in comparison to temperature contrasts of the other models."; a still quit clumsy sentence and also counter-intuitive, as in case BESM shows a more positive lapse rate feedback than the ensemble, this should imply that the warming is relatively larger at the surface and relatively weaker at the upper troposphere, resulting in an in fact more strongly increased (!) vertical temperature gradient. That seems to contradict your statement. Do you agree? Please, reformulate accordingly.**
Reply: we agree with the suggestion. Indeed, the BESM warming is relatively larger at the surface and relatively weaker at the upper troposphere, as indicated by the comment. (p. 9 l. 6)

**9) Major point! The interpretations starting at p. 10, l. 4, with respect to Fig. 6, still lack clarity and also inherent consistency. First, it is not obvious how the term 'cloud mask' is to interpret physically, as it is not mentioned when the 'adjusted cloud radiative effect', ΔCREa, is introduced by Eq. (5). As a consequence, the link between the analytic explanation in the penultimate paragraph of 4.2 and the discussion of physical parameters in the last paragraph remains obscure. In particular (my second main point) it is not clear to me how (l. 15) "Because of the increase in total cloud fraction, a negative SW CRE appears …" can be reconciled with the SW cloud feedback in Fig. 5 (mid right panel) being obviously positive in those very regions. Does the lower troposphere cloud response in BESM compete with the sea ice retreat differently than in the CMIP ensemble? While a full explanation is not required in the framework of a GMD paper, still you should try to explain the parameter inter-relations in a consistent and lucid way. Especially, because the concluding section resumes this point to emphasize it as a main part of the paper.**
Reply: The physical interpretation was added after the Eq. 5 introduction. We think that it can make the result discussion clear (p. 7 l. 14). About the second major concern, which claims that the SW CRE is positive, in fact it is negative (see the first row of the Figure 6). This SW CRE is not adjusted, it is as described by Cess et al. (1989). The conciliation between figures 5 and 6 is the main point of the Figure 6 analysis. The SW cloud feedback was divided into their contributors: the SW CRE; the SW cloud masking for the water vapor feedback; and, the cloud masking for the albedo feedback. Although BESM has a negative values over Arctic and Sub-antarctic regions (as other models), it is corrected (adjusted) by the cloud masking for the non-cloud feedbacks, resulting in a very positive SW cloud feedback over those regions, as previous shown in Figure 5.

**10) p. 13 l. 20: "… increase in cloud cover … which increased the outgoing SW radiation at the TOA." ; see previous point 9! In Fig. 5 the SW cloud feedback in that latitude range is positive, which suggests that more SW energy remains in the climate system, thus indicating less reflection. By the way, "outgoing SW radiation" ought to be replaced by "reflected solar radiation".**
Reply: the answer to the item 9 can be used to item 10. The "outgoing SW radiation" term was replaced by "reflected solar radiation". (p. 13 l. 21)

**11) p. 13 l. 3: "… overcome by the albedo feedback cloud mask..."; also related to major point 9. This explanation is not suggestive and not adequately pointed out in a physical sense.**
Reply: the answer to the item 9 can be used to this item.

**12) p. 14 l. 2: "in this sence it has contributed …" You obviously refer to that part of the manuscript I have addressed in my general points 9-11. Evidently, I do not agree to this claim as not convincing argument for a "better understanding" has been provided. Nor do I demant or expect, that such a contribution to better understanding is given in the framework of a GMD paper. Hence, the statement could simply be discarded, or modified to something like "… the analysis methods used here have the potential to explain remaining process uncertainties causing inter-model spread in the cloud feedback in future work."**
Reply: we agree and the sentence was included to the manuscript. (p. 14 l. 3)

**Language and Technical Remarks:**

Almost all suggestions were accepted. The exceptions are listed (by their numbers) below:

3) The suggestion modifies the original meaning. Thus, we rewrite the sentence to make it clear.

9) It is right. Just the last 150 years of the PiControl run have been used.

34) Even though it is the same subject (p. 9 l. 2 and l. 11 of the original version), the second sentence is localizing in which latitude BESM is an outlier model. So, we decided do not discard such sentence.

References

[revised manuscript text omitted]